# Mutation analysis links angioimmunoblastic T-cell lymphoma to clonal hematopoiesis and smoking

**Shuhua Cheng[1], Wei Zhang[2], Giorgio Inghirami[1], Wayne Tam[1]\***

[1]Department of Pathology and Laboratory Medicine, Weill Cornell Medicine, New York, United States; [2]Genomics Resources Core Facility, Weill Cornell Medicine, New York, United States

## Abstract

**Background:** Although advance has been made in understanding the pathogenesis of mature T-cell neoplasms, the initiation and progression of angioimmunoblastic T-cell lymphoma (AITL) and peripheral T-cell lymphoma, not otherwise specified (PTCL-NOS), remain poorly understood. A subset of AITL/PTCL-NOS patients develop concomitant hematologic neoplasms (CHN), and a biomarker to predict this risk is lacking.

**Methods:** We generated and analyzed the mutation profiles through 537-gene targeted sequencing of the primary tumors and matched bone marrow/peripheral blood samples in 25 patients with AITL and two with PTCL-NOS.

**Results:** Clonal hematopoiesis (CH)-associated genomic alterations, found in 70.4% of the AITL/PTCL-NOS patients, were shared among CH and T-cell lymphoma, as well as concomitant myeloid neoplasms or diffuse large B-cell lymphoma (DLBCL) that developed before or after AITL. Aberrant AID/APOBEC activity-associated and tobacco smoking-associated mutational signatures were respectively enriched in the early CH-associated mutations and late non-CH-associated mutations during AITL/PTCL-NOS development. Moreover, analysis showed that the presence of CH harboring ≥2 pathogenic TET2 variants with ≥15% of allele burden conferred higher risk for CHN (p=0.0006, hazard ratio = 14.01, positive predictive value = 88.9%, negative predictive value = 92.1%).

**Conclusions:** We provided genetic evidence that AITL/PTCL-NOS, CH, and CHN can frequently arise from common mutated hematopoietic precursor clones. Our data also suggests smoking exposure as a potential risk factor for AITL/PTCL-NOS progression. These findings provide insights into the cell origin and etiology of AITL and PTCL-NOS and provide a novel stratification biomarker for CHN risk in AITL patients.

**Funding:** R01 grant (CA194547) from the National Cancer Institute to WT.

**\*For correspondence:** wtam@med.cornell.edu

**Competing interest:** The authors declare that no competing interests exist.

## Introduction

Peripheral T-cell lymphoma (PTCL) is a heterogenous group of lymphoid tumors and encompass peripheral T-cell lymphoma, not otherwise specified (PTCL-NOS), angioimmunoblastic T-cell lymphoma (AITL), and several other entities of T-cell lymphoma (*Swerdlow et al., 2017*), likely driven by an array of recurrent genomic defects (*Fiore et al., 2020a*). Except for PTCL-NOS, AITL is the most common subtype of PTCL (21–36.1%) (*Chiba and Sakata-Yanagimoto, 2020*; *de Leval et al., 2015*) and is believed to arise from a subset of peripheral mature CD4+ T-cells corresponding to T-follicular helper (TFH) cells, characterized immunophenotypically by expression of a set of cellular markers like PD1, CXCR5, BCL-6, CD10, CXCL13, and ICOS-1 (*Attygalle et al., 2002*; *Chiba and Sakata-Yanagimoto,*

*2020*; *Dupuis et al., 2006*; *Marafioti et al., 2010*; *Mourad et al., 2008*). Morphologically, AITL is typically characterized by a polymorphous lymphoid infiltrate with a proliferation of medium-sized tumor cells with clear cytoplasm (clear cell immunoblasts), associated with prominent proliferation of high endothelial venules and follicular dendritic cells. A subset of PTCL, termed PTCL with TFH phenotype in the updated WHO classification, is thought to be also derived from TFH and may be biologically related to AITL, sharing some clinical-pathological features with the latter (*Swerdlow et al., 2017*). Although progress has been made in understanding AITL pathogenesis and developing new treatment (*Lemonnier et al., 2018*), AITL remains as an aggressive lymphoid tumor, with low estimated rates of overall and failure-free survival at 5 years (33% and 18%, respectively) (*Federico et al., 2013*). To develop more effective therapeutic agents against AITL and PTCL in general, with TFH phenotype further understanding of the molecular pathogenic mechanisms of AITL is needed.

Genetically, AITL is characterized by a number of genomic mutations in *TET2*, *RHOA*, *DNMT3A*, and *IDH2* (*Fiore et al., 2020a*; *Nakamoto-Matsubara et al., 2014*; *Odejide et al., 2014*; *Sakata-Yanagimoto et al., 2014*; *Yoo et al., 2014*). Cell-intrinsic and/or -extrinsic factors that facilitate the accumulation of these AITL-related mutations remain unclear. Mutations in *TET2* and *DNMT3A* are also frequently associated with myeloid malignancies, including acute myeloid leukemia (AML), myeloproliferative neoplasms (MPN), and myelodysplastic/myeloproliferative neoplasms (MDS/MPN). *TET2* and *DNMT3A* are also the most commonly mutated genes associated with clonal hematopoiesis (CH) in healthy adults, especially those over 60 years of age. CH has been shown to be an aging-related process characterized by the clonal expansion of hematopoietic cells harboring one or more somatic mutations as a result of selective advantage in the hematopoietic stem and progenitor cells due to enhanced self-renewal and inhibition of differentiation (*Challen and Goodell, 2020*; *Gondek and DeZern, 2020*; *Jaiswal and Ebert, 2019*; *Steensma and Ebert, 2020*). It has been noted that myeloid and lymphoid malignancies may co-occur in the same patients (*Holst et al., 2020*). Considering the similarities in genomic mutation profiles of AITL, myeloid malignancies, and CH, it has been postulated that there may be a biological link between these entities. To test this, the current study implemented next-generation sequencing (NGS) approach to analyze neoplastic T-cells and paired bone marrow/peripheral blood (BM/PB) specimens from a cohort of 27 patients with AITL or PTCL-NOS, and explored the potential of using the genomic findings from this study to shed light into the etiology of AITL development and to predict clinical progression related to development of second hematologic neoplasms, which is one of the clinical challenges in the clinical management of these cancer patients.

# Materials and methods

**Key resources table**

| Reagent type (species) or resource | Designation | Source or reference | Identifiers | Additional information |
|---|---|---|---|---|
| Commercial assay or kit | KAPA HyperPlus Kit | Roche | Catalog # 07962363001 | |
| Commercial assay or kit | Twist Hybridization and Wash Kit | Twist Bioscience | Catalog # 101025 | |
| Commercial assay or kit | Lymphoma pilot (16X ), lot 3020 | Twist Bioscience | Catalog # 3020 | |
| Commercial assay or kit | HiSeq 3000/4000 SBS Kit | Illumina | Catalog # FC-410-1001 | |
| Commercial assay or kit | HiSeq 3000/4000 PE Cluster Kit | Illumina | Catalog # PE-410-1001 | |
| Other | HiSeq 4000 System | Illumina | RRID:SCR_016386 | |
| Software, algorithm | NextGENe | SoftGenetics, LLC | RRID:SCR_011859 Catalog # NG001 version 2.4.2.3 | |
| Software, algorithm | MutSignature | *Fantini, 2021*, https://github.com/dami82/mutSignatures | Version 2.1.1 | |

*Continued on next page*

*Continued*

| Reagent type (species) or resource | Designation | Source or reference | Identifiers | Additional information |
|---|---|---|---|---|
| Software, algorithm | Maftools | *Mayakonda, 2021*, https://github.com/PoisonAlien/maftools | Version 2.4.12 | |
| Software, algorithm | R base package | https://www.r-project.org/ | RRID:SCR_002394 version 4.0.2 | |
| Software, algorithm | Prism | GraphPad | RRID:SCR_002798 version 5 | |

## Patients and study samples

All tissue samples (27 lymph node [LN] tissue specimens, 27 BM aspirate/PB samples) were collected from 25 AITL or 2 PTCL NOS patients who were diagnosed or confirmed from June 2010 to December 2019 following World Health Organization classification criteria by attending hematopathologists at NYP/Weill Cornell Medical Center, and clinical Information was obtained from electronic clinical records. Of these 27 study cases, 4 were initially diagnosed with PTCL with THF phenotype (*Supplementary file 1*) and were included in the AITL group based on their similar clinical and molecular features as recently proposed by WHO (*Swerdlow et al., 2017*). The two PTCL-NOS cases do not show significant expression of TFH-associated markers based on immunohistology. The clinical-pathological features of these two cases are as follows: patient #2: mesenterial lymphadenopathy found on CT scan during work-up for renal transplant, no morphological features of AITL, predominantly small cells. The T-cells were positive for CD2, CD3, CD5, CD7, CD4, negative for CD8, CD10, BCL6, and PD-1, diagnosed as PTCL-NOS. Patient #18: abdominal and cervical lymphadenopathy, large pleomorphic cells. The T-cells were positive for CD2, CD3, CD5, CD8, TIA-1, granzyme B, TCR alpha-beta, negative for CD7, CD4, CD10, CD56, CD57. diagnosed as PTCL-NOS, with cytotoxic phenotype.

For tumor burden (TB) estimate in the BM/PB samples, a complementary strategy was implemented due to limitations of each histological or molecular methods. Histological examination has a low sensitivity and AITL cells might lack distinct morphological or immunophenotypic features in the BM/PB samples, potentially leading to false negativity in histological or immunophenotyping estimation in some cases (e.g., patient #1, #5, #12). To avoid these potential pitfalls, besides considering morphological findings, the TB estimate was also based on more objective and sensitive immunophenotypic findings (flow cytometry, Flow). If flow was negative and T-cell receptor gamma gene rearrangement (TCRG) was positive, we estimated TB based on the analytic sensitivity of the TCRG assay, which is about 1–5%. If both Flow and TCRG were negative, the variant allele frequencies (VAFs) of the T-cell lymphoma-associated variants like RHOA p.V17A would be used for estimation by comparison (e.g., the PB or BM samples from patient #1, #22, *Supplementary file 1*).

This study was conducted in accordance with the Declaration of Helsinki regulations of the protocols approved by the Institutional Review Board of Weill Cornell Medicine, New York, USA (#0107004999). Written consent for use of the samples for research was obtained from patients or their guardians.

Genomic DNA was extracted from LN tissue and BM or PBMC cell pellets following the manufacturer's instructions (QIAamp DNA Mini Kit, Qiagen, Germantown). DNA samples and sequencing libraries used in targeting sequencing as described below were quantitated by Tape Station (Agilent Technologies, Santa Clara) and Qubit (Thermo Fisher Scientific, Singapore).

## T-cell targeted sequencing

A 537-gene targeted sequencing panel (*Supplementary file 2*), based on hybridization capture method for sequencing library construction and selection, was designed to investigate the genomic profile of the primary tumors and the BM/PB tissues (*Fiore et al., 2020b*). The genomic regions covered by sequencing include coding exons and splice sites of these genes (target region: ~3.2 Mb) that were reported being recurrently mutated (>2) in mature T-cell neoplasms, as well as genomic regions corresponding to recurrent translocations. Using an input of genomic DNA of at least 100 ng isolated from frozen tissues or FFPE samples, the NGS libraries were constructed using the KAPA Hyperplus Kit (Roche, Basel, Switzerland), and hybrid selection was performed with the probes from the customized Twist Library Prep Kit (Twist Biosciences, San Francisco, CA), according to the

manufacturer's protocols. Multiplexed libraries were sequenced using 150 bp paired end HiSeq 4000 sequencers (Illumina, San Diego, CA).

NextGENe software (SoftGenetics, State College, PA) was used to perform bioinformatic analysis (SNV and INDEL variant calls) with standard settings recommended by the manufacturer. Specifically, the pipeline settings are as follows: read quality reject or trimming (Q score <20, ≥3 bases with Q score ≤ 10), Allowable Mismatched Bases (0), Allowable Ambiguous Alignments (50), Seed (40 bases), Move Step (15 bases), Allowable Alignments (100), Matching Base Percentage ≥ 97.0, Detect Large Indels (TRUE), Sequence Range Checked (FALSE), Hide Unmatched Ends (TRUE), Except for Homozygous (FALSE), Mutation Filter Use Original (TRUE), Variation Mutation Percentage ≤ 5.00, Variation SNP Allele ≤ 5 Counts, Variation Total Coverage ≤ 50, Indels Mutation Percentage ≤ 5.00, Indels SNP Allele ≤ 5 Counts, Indels Total Coverage ≤ 50, HomoIndels Mutation Percentage ≤ 5.00, HomoIndels SNP Allele ≤ 5 Counts, HomoIndels Total Coverage ≤ 50, Perform in-read phasing (TRUE), Max gap between two variants 1 (0–3), Phaseable reads percentage ≥ 50.00, Max Phase alleles count (2), Load Assembled Result Files (FALSE), Load Sage Data (FALSE), Load Paired Reads (TRUE), Min Pair End Gap (0), Max Pair End Gap (200), Save Matched Reads (FALSE), Highlight Anchor Sequence (FALSE), Ambiguous Gain/Loss (FALSE), and Detect Structure Variations (FALSE). Additionally, cutoff values of the post-alignment filter parameters for the VAF, population frequencies, strand balance ratio relative to counts measuring for strand bias, and function prediction were set at 5%, 0.01%, 1:5, and >2, respectively. Human_v37p10_dbsnp135 (hg19) was used as human reference genome for alignment.

### Myeloid NGS panel

Targeted enrichment of 45 genes recurrently mutated in myeloid malignancies (*Supplementary file 2*) was performed using the Thunderstorm system with a customized primer panel (*Cheng et al., 2017*). The primers target coding exons of the genes, leading to a total of 726 amplicons. Libraries were prepared by microdroplet-based PCR target enrichment method from DNA, followed by sequencing using the Illumina MiSeq yielding 260 bp paired end reads. Sequencing data were analyzed and reported with a customized analytical pipeline. This NGS panel testing was performed in a clinical lab CLIA-certified and accredited by the College of American Pathologists.

### Data analysis

Most of the data analysis were conducted with GraphPad/Prism 5 software and various R packages, including base packages, ggplot2, ComplexHeatmap, Maftools, and MutSignatures. The survival comparison was analyzed using Kaplan–Meier curves (log-rank test, significance defined as p<0.05). z-test was conducted with an online calculator (https://www.socscistatistics.com/tests/ztest/default2.aspx).

## Results

### Mutation profiling of AITL/PTCL-NOS and matched BM/PB supports a potential origin of AITL/PTCL-NOS from mutated hematopoietic precursors associated with CH

For mutation profiling, we sequenced 27 pairs of AITL or PTCL-NOS samples using a 537-gene targeted NGS panel that covered recurrently mutated genes associated with T-cell lymphomas (*Fiore et al., 2020b*). Of the genomic regions targeted by the panel, 90% had a coverage depth of >1000. Those sequenced samples included 27 diagnostic LN specimens from patients with AITL (n = 25) or PTCL-NOS (n = 2) and their corresponding BM (n = 21) or PB samples (n = 6) from our archived specimens (hereafter denoted as AITL/PTCL-NOS). The overall genomic and pathological findings showed that of the 27 BM/PB samples, 10 had no detectable involvement by AITL or PTCL-NOS (37%), while 17 were involved by the neoplastic T-cells (63%) of variable abundance (*Supplementary file 1*). One BM sample showed concomitant diagnostic involvement by an MPN (patient #20) (*Supplementary file 1*).

The genomic alterations found in the matched BM/PB can be due to (1) BM/PB involvement by AITL/PTCL-NOS, (2) CH, or both (1) and (2). To accurately characterize the mutation spectrum in the BM/PB, we distinguished the CH-associated mutations from those attributed to the BM/PB involvement by the T neoplastic cells according to an algorithm as described in the Methods

section. Briefly, the TB was estimated for each of the BM/PB specimens involved by the lymphomas (16 AITL and 1 PTCL-NOS) based on their histological, immunophenotyping, and T-cell receptor gamma (*TCRG*) gene rearrangement findings (*Supplementary file 1*), and compared to the VAFs of the somatic alterations of T-cell lymphoma-associated genes, for example, *RHOA* p.G17V, a molecular characteristic of AITL (*Nakamoto-Matsubara et al., 2014*; *Sakata-Yanagimoto et al., 2014*; *Tiacci et al., 2018*; *Yoo et al., 2014*). The AITL/PTCL-NOS-related variants present in the BM/PBs are highlighted in red in *Supplementary file 3*, and their VAFs (VAF$^{inv}$) were found to be ~1% on average (median, ~1% ; range: 0.1–6%) (*Figure 1—figure supplement 1*). The variants whose VAFs could not be attributed to AITL/PTCL-NOS involvement alone in the BM/PB, or the variants detected in BM/PB uninvolved by lymphoma were presumed to correspond to variants related to CH, which was confirmed by the presence of these CH-associated mutations in purified neutrophils in the PB of one patient (patient #24, *Supplementary file 3* and *Figure 1—figure supplement 2*). Compared to the AITL/PTCL-NOS-related variants, the VAFs of the CH-associated variants (VAF$^{CH}$) were significantly higher (p<0.0001), ranging from 0.2% to 51.5% with a mean of 22.5% (median: 16.5%), approximately 21.2 times higher than the VAF$^{inv}$ on average (*Figure 1—figure supplement 1C*).

We identified in the matched BM/PB specimens 44 variants from 14 genes, excluding the variants attributed to the BM/PB involvement by AITL/PTCL-NOS as described above (*Figure 1B*). These alterations included 17 missense (38.64%) and 12 nonsense (27.27%) SNVs, 6 frameshift (13.64%) deletions, 7 frameshift (15.91%), and 1 in-frame insertions (*Figure 1*, *Figure 1—figure supplement 3*, *Supplementary file 4*). Among these 44 somatic mutations, 37 mutations, identified in 19 of the 27 (70.4%) cases, were shared with those found in the primary lymphoma, and 7 were BM/PB specific (*Figure 1*, *Supplementary file 3*). The recurrent shared mutations were primarily restricted to *TET2* (74% of the cases) and *DNMT3A* (37% of the cases), consistent with the top CH-associated genes previously reported (1C ) (*Genovese et al., 2014*). All but three (patient #5, #20, #24) of the cases with CH-associated mutations in the BM/PB did not have a dx of an overt myeloid neoplasm. These CH-associated mutations presumably were acquired very early in the common ancestral hematopoietic precursor cells from which both the myeloid and T-cell lineages are derived. Consequently, we defined these shared CH-associated variants as early mutations as seen below. In eight patients (patient #7, #9, #11, #12, #17, #21, #23, #26) (29.6%), no CH-associated mutations were detected in the BM or PB.

In the 27 diagnostic LN samples, we identified a total of 102 non-synonymous somatic mutations in 37 genes with a median of ~3 variants per sample, including 62 missense (60.78%) and 20 nonsense (19.61%) single-nucleotide variants (SNVs), 1 in-frame (0.98%) and 9 frameshift (8.8%) deletions, and 1 in-frame and 7 frameshift (6.9%) insertions (*Figure 1B*, *Supplementary file 4*, *Figure 1—figure supplement 4*). Of these 102 mutations, 37 were associated and shared with CH (*Figure 1B and C*). In more than half of the T lymphoma cases, not only could we detect early CH-associated mutations, we also identified 65 mutations that are likely acquired during the later stage of AITL/PTCL-NOS development (referred as late mutations hereinafter) (*Figure 1B and C*, *Figure 1—figure supplements 2 and 4*). The recurrent late mutations were limited to several oncogenes and tumor suppressor genes, including the well-known driver genes like *RHOA* (67% of the cases), *TET2* (48%), *IDH2* (33%), *PLCG1*(10%), *TP53*(10%), *VAV1* (10%), and are characterized both by the absence of *DNMT3A* mutations (*Figure 1C*) and by the enrichment of missense mutations, which were increased from 36.1% in the CH-associated mutations to 75.2% in the late mutations (proportion test, p-value<0.0001; *Figure 1D*). The mutations in *IDH2*, *PLCG1*, and *TP53* were found exclusively as late mutations and not CH-associated mutations (*Figure 1C*).

*Figure 1A* shows four representative AITL cases and two PTCL-NOS cases with their matched BM/PB, where red dots indicate the CH-associated variants present in both the primary lymphoma and BM/PB, and black dots represent the variants associated with AITL/PTCL-NOS (also highlighted with rectangles). A detailed description of these illustrative cases is provided in Appendix 1. There are a couple of notable findings: first, more than one CH clone can be present in the BM, and their clonal representations in the BM may not reflect those in the lymphoma, as seen in the *DNMT3A*-mutated clones in patient #4. These results suggest that the same *DNMT3A* mutation can have differential effect depending on the cell lineage affected. Second, findings in patient #29 raise the possibility that besides the neoplastic T-cells, reactive lymphocytes in these two cases might also harbor the CH-associated mutations.

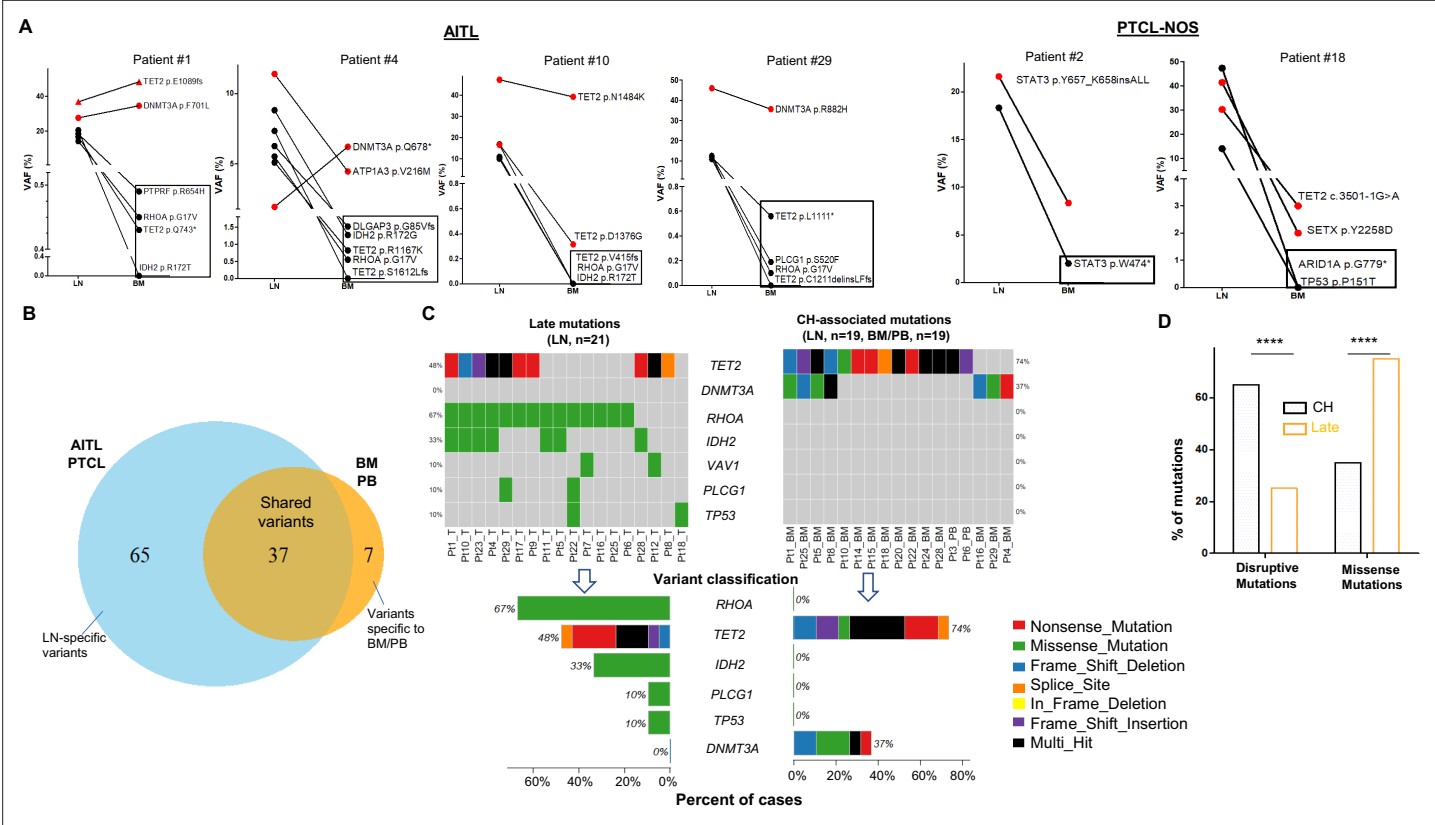

**Figure 1.** Analysis of genomic alterations by target sequencing panel for primary lymphomas and paired bone marrow/peripheral blood (BM/PB) in patients with angioimmunoblastic T-cell lymphoma (AITL) and peripheral T-cell lymphoma, not otherwise specified (PTCL-NOS). (**A**) Presence of clonal hematopoiesis (CH) in patients with AITL and PTCL-NOS. Dot plots showing the detected variants and their variant allele frequencies (VAFs) in the AITL and PTCL-NOS (lymph node [LN]) and their matched BM/PB in representative AITL and PTCL-NOS cases with CH. The black circles indicate variants specific to the lymphomas, and the variants shared between the primary lymphomas and CH are highlighted in red. The variants attributed to lymphoma only are boxed. Additional detailed descriptions of these illustrative cases are provided in Appendix 1. (**B**) Venn diagram showing the distribution of the shared, lymphoma or BM/PB-specific variants identified in the diagnostic LN and paired BM/PB samples. The shared variants are defined as variants identified in both the primary lymphoma and the BM/PB, the latter as CH-related variants. The variants predicted to be due only to lymphoma involvement in BM/PB have been excluded (see also *Figure 1—figure supplement 1A* for the distribution of all variants). (**C**) Summary of the CH-associated mutations in the BM/PB and LN, and the mutations postulated to accumulate at a later stage of lymphoma development (late mutations). The CH-associated mutations are shared between the primary lymphomas and the BM/PB and can be considered as early lesions in AITL/PTCL. The heatmaps show the top recurrent mutations in both categories. Stacked bar plots show the type of variants and the mutation frequency (relative to our cohort) for each of the major mutated genes in the LN and BM/PB samples. Pt: patient; T: tumor. (**D**) Comparison of the distribution of disruptive and missense mutations in the CH-associated and late mutations. Statistical significance was determined by (**D**) z test measuring proportion difference. *p<0.05; **p<0.01; ***p<0.001; ****p,0.0001; NS, not significant. P-Value<0.05 is considered statistically significant.

The online version of this article includes the following figure supplement(s) for figure 1:

**Figure supplement 1.** Angioimmunoblastic T-cell lymphoma (AITL) variants involving bone marrow/peripheral blood (BM/PB).

**Figure supplement 2.** Comparison of variant allele frequencies (VAFs) of the variants found in paired angioimmunoblastic T-cell lymphoma (AITL) and bone marrow/peripheral blood (BM/PB) samples.

**Figure supplement 3.** Overall and clonal hematopoiesis (CH)-related genomic alterations in the matched bone marrow/peripheral blood (BM/PB) samples.

**Figure supplement 4.** Overall and lymphoma-specific genomic alterations in primary angioimmunoblastic T-cell lymphoma/peripheral T-cell lymphoma, not otherwise specified (AITL/PTCL-NOS).

Our results support a tumor model in which AITL/PTCL-NOS emerges from mutated and expanded HP clones that are associated with CH in the BM as well as serving as the lymphoma precursors. The latter often acquires additional missense mutations during the course of development to frank lymphomas.

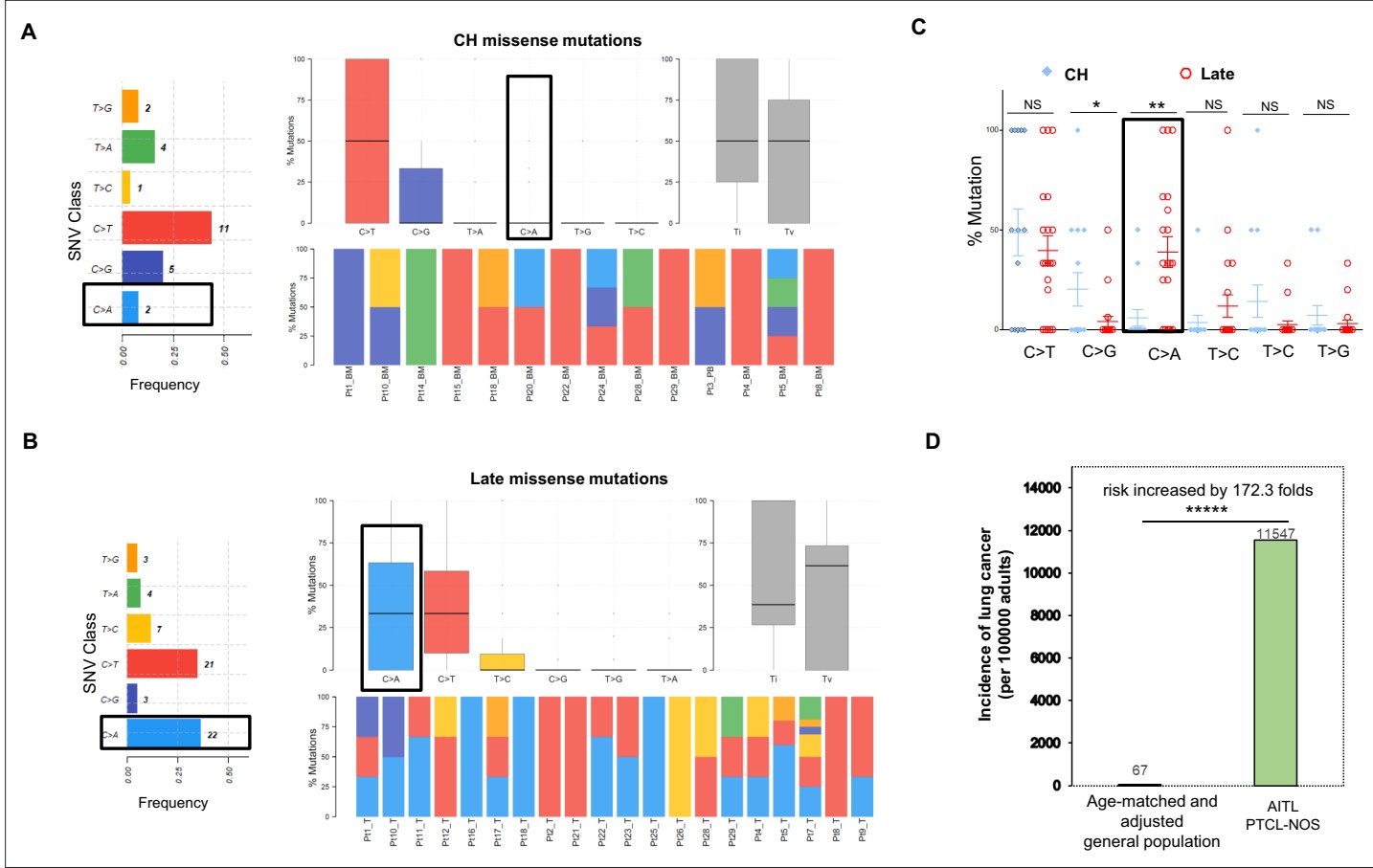

**Figure 2.** Late mutations in angioimmunoblastic T-cell lymphoma/peripheral T-cell lymphoma, not otherwise specified (AITL/PTCL-NOS) are enriched for C>A transversion substitutions possibly associated with smoking. Transition and transverse (Titv) plot showing overall distribution of the six types of substitutions in the clonal hematopoiesis (CH) (**A**) and late (**B**) missense mutations acquired during AITL/PTCL development, as well as fraction of these substitutions in each sample. The median is indicated by a horizontal line. Bar plot on the left showing single-nucleotide variant (SNV) classes and fraction of each substitution class among all missense mutations. (**C**) Side-by-side comparison of transition and transversion base substitutions acquired between the early CH-associated and late mutations. (**D**) Bar plot comparing the incidence rate of lung cancer between two age-matched/-adjusted populations indicated. Statistical significance was determined by (**C, D**) z test. *p<0.05; **p<0.01; ***p<0.001; ****p<0.00001; NS, not significant.

The online version of this article includes the following figure supplement(s) for figure 2:

**Figure supplement 1.** Comparison two de novo mutational signatures from clonal hematopoiesis (CH)-associated angioimmunoblastic T-cell lymphoma (AITL) mutations with COSMIC Signatures.

**Figure supplement 2.** Similarity of two de novo mutational signatures from angioimmunoblastic T-cell lymphoma (AITL) late mutations with COSMIC Signatures.

**Figure supplement 3.** Similarity of two de novo mutational signatures from the published T-follicular helper-peripheral T-cell lymphoma (TFH-PTCL) mutation dataset with COSMIC Signatures.

**Figure supplement 4.** Comparing de novo signatures extracted from angioimmunoblastic T-cell lymphoma (AITL) late mutations and published T-follicular helper-peripheral T-cell lymphoma (TFH-PTCL) mutations.

**Figure supplement 5.** Comparing de novo signatures extracted from angioimmunoblastic T-cell lymphoma (AITL) late mutations in the non-smokers to COSMIC Signatures.

## Late mutations in AITL/PTCL-NOS are enriched for C>A transversion substitutions possibly associated with smoking

We investigated whether there might be an etiological difference between the CH-related mutations and the late mutations by analyzing mutational signatures.

For the CH-associated mutations, overall transition (Ti) and transverse (Tv) substitution rates are comparable (Ti vs. Tv, median, 50% vs. 50%, mean, 54.76% vs. 45.24%, p-value>0.05; *Figure 2A*). At

the base substitution level, C>T is found most frequently (44%), followed by C>G (20%; *Figure 2A*). We extracted two major de novo mutational signatures (CH_Sign.01 and 02, *Figure 2—figure supplement 1A*) from the CH-related mutations by MutSignatures (*Fantini et al., 2020*). CH_Sign.01 is characterized by the enriched C>T substitutions at the trinucleotide motif Tp**C**pA (mutated base presented as bold), and CH_Sign.02 is enriched with C>T at Cp**C**pA/Gp**C**pG plus the increased C to G substitutions at Tp**C**pG. A cosine correlation similarity (CCS) was used to evaluate closeness between the CH de novo and COSMIC (SBS30, version 2) signatures. CCS, measured as 1 - cosine distance, ranges from 0 to 1. 0 denotes completely different mutational signatures and 1 signifies identical signatures. As shown in *Figure 2—figure supplement 1B*, CH_Sign.01 demonstrates the best match with COSMIC Signature 2 (CCS = 0.65), which is associated with activity of the AID/APOBEC family of cytidine deaminases (*Alexandrov et al., 2013*). The characterized trinucleotide change in CHSign.01, Tp**C**pA to Tp**T**pA, is also the hallmark of COSMIC Signature 2. Analysis of the CH mutations as the consequence of each mutational signature per sample showed that the activity of CH_Sign.01 dominated in 75% (15/20) of the AITL/PTCL-NOS samples (*Figure 2—figure supplement 1C*), indicating a potential major role of the AID/APOBEC family of cytidine deaminases in the generation of CH-associated mutations in AITL. Another CH signature Sign.02, active in 55% (11/20) of the samples, is closest to COSMIC Signature 15 (CCS ≈ 0.50), reportedly attributed to defective DNA mismatch repair.

In the late mutations (LM), Ti and Tv are also not significantly different (Ti vs. Tv, median, 38.53% vs. 61.43%, mean, 49.31% vs. 50.68%, p-value>0.05). At the base substitution level, however, besides C>T (35%), C>A emerges as one of the predominant mutant forms (36.7%; *Figure 2B*). On a case basis, C>A substitutions are enriched in late mutations compared to CH mutations (mean, 38.8% vs. 5.95% ; median, 33.3% vs. 0% ; t test, p-value=0.0024), and the fraction of the cases with C>A substitution in the late mutations is 4–5 times that with C>A in the CH-associated mutations (67% vs. 16%; *Figure 2C*). Signature analysis identified two de novo mutation signatures (LM_Sign.01 and LM_Sign.02, *Figure 2—figure supplement 2A*). LM_Sign.01 is enriched with the Tp**C**pC to Tp**A**pC mutation, which is one of the main trinucleotide motifs with the C>A base substitutions identified in COSMIC Signature 4 attributed to the smoking-induced mutational process (*Alexandrov et al., 2013*). Consistent with this, analysis of cosine similarity revealed that the de novo signature LM_Sign.01 had the closest match with COSMIC Signature 4 (CCS ≈ 0.5), followed by Signature 24 (CCS = 0.4, associated with aflatoxin), highly active in 75% of the tumor samples as evidenced by 50% or more of the mutations in each sample (1–8 variants, mean mutation number = 1.74) as the result of the LM_Sign.01 signature activity (*Figure 2—figure supplement 2B and C*). Like COSMIC Signature 4, LM_Sign.01 exhibited transcriptional strand bias for the C>A substitutions where the mutation of C on the forward strand (C>A, n = 17) exceeded the mutation of G on the reverse strand (G>T, n = 5) by 2.4-folds. Therefore, these findings suggest a potential causative link between smoking or secondhand smoking (SHS) and AITL development by acquisition of additional driver mutations in the pathogenesis of AITL.

The second extracted signature LM_Sign.02, active in all the tumor samples, was enriched with the C>T change at the triplex motif Cp**C**pC and showed the closest match with multiple COSMIC Signatures, including #23 (CCS = 0.55, etiologically unknown), #11 (CCS = 0.49, associated with treatment of alkylating agents), and #19 (CCS = 0.48, etiologically unknown) (*Figure 2—figure supplement 3*). This suggests that, besides smoking-induced tumorigenesis, other mutational processes might also contribute to acquisition of driver mutations.

We analyzed a published targeted genomic sequencing dataset derived from 44 patients diagnosed as PTCL with TFH phenotype (Kyoto cohort)(*Watatani et al., 2019*). This analysis unfolded two major de novo mutational signatures (Kyoto_Sign.01 and Kyoto_Sign.02). As observed in LM_Sign.01 described above, Kyoto_Sign.01 was enriched with the Tp**C**pC to Tp**A**pC nucleotide substitution, which largely occurred in *RHOA* and *TET2* genes. The Kyoto_Sign.01 signature also exhibited the transcriptional strand bias, with the C>A mutations on the forward strand exceeding those on the reverse strand by 1.9-folds (*Figure 2—figure supplement 3A*). Further analysis showed that it had the closest match with COSMIC Signature 4 (CCS = 0.52) and was highly active in about 32 out of 44 TFH-PTCL cases (73%; *Figure 2—figure supplement 3B and C*). Cosine similarity reveals that Kyoto_Sign.01 was almost identical with LM_Sign.01 (CCS ≈ 0.9, *Figure 2—figure supplement 4*), validating the above signatures and the potential link to cigarette smoke.

Identification of the potential cigarette smoke-associated mutation signature in the late mutations raises the possibility that patients with AITL/PTCL-NOS might have a higher risk for lung cancer or other smoking-associated cancers. To test this hypothesis, we compared the annual incidence rate of lung cancer in patients with AITL/PTCL-NOS (the current study cohort, n = 28, including one additional PTCL-NOS case without matched PB/BM control, patient #27, see *Supplementary file 1*), and age-matched/adjusted general US population (n = 186,293,423, estimated) as control group. The data for the control group covering 12 years (2006–2017) in the 30–85+ age groups were downloaded from the CDC website: https://gis.cdc.gov/Cancer/USCS/DataViz.html (data not available after 2017). The patients in the current study cohort were diagnosed with AITL or PTCL-NOS during 12 years (2008–2019, Weill Cornell Medicine) with an age range of 33–84 years old (median, 65; mean, 62). The weights used in the age adjustment of the data are the proportion of the 2010 US standard population within each age group. The incidence rates of lung cancer were calculated according to the following formula: new lung cancer/age-matched population * 100,000 * weight for the age adjustment. Analysis shows that the incidence rate of lung cancer in AITL/PTCL-NOS is 172.3 times higher than that in the age-matched general population (11,547 vs. 67, p<0.00001; *Figure 2D*), further demonstrating that AITL/PTCL-NOS, like lung cancer, might be causatively linked to smoking or involuntary smoking.

Medical records showed that 7 (26.9%) of the AITL/PTCL-NOS patients in our cohort were smokers (one passive smoking), 19 (73.1%) non-smokers, and 2 no records. Patient #27, one of the three patients with synchronous lung cancers, was documented with smoking history of one pack per day before onset of PTCL-NOS, and the other two lung cancer patients were non-smokers. No significant difference was detected in the C>A or overall Signature 4 mutation burden per sample between evaluable smokers and non-smokers (average number of the C>A mutations: 1 vs. 0.92, p=0.61).

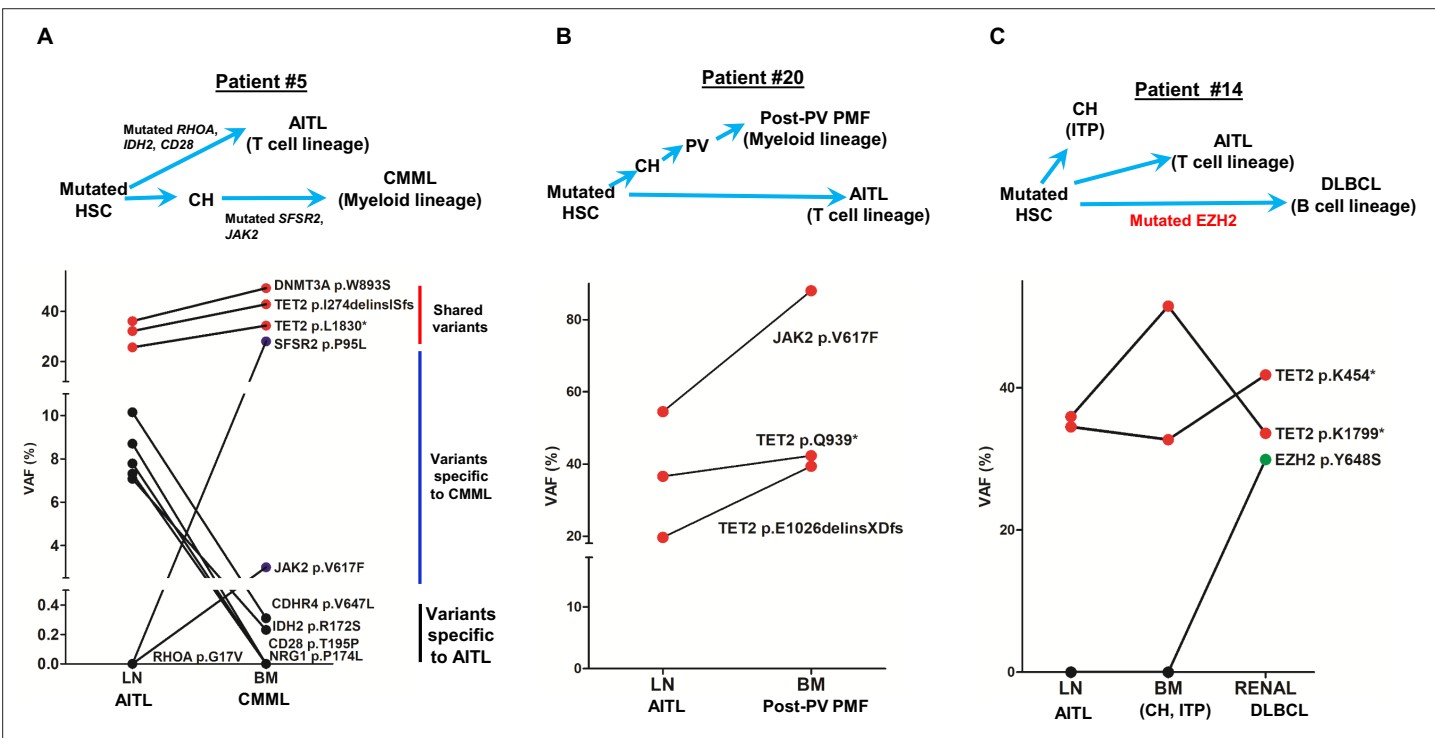

**Figure 3.** Angioimmunoblastic T-cell lymphoma (AITL) and concomitant hematologic neoplasms develop from common mutated hematopoietic precursor cells. (A–C) Dot plots comparing variant allele frequencies (VAFs) of the mutations identified in the AITL and the concomitant hematologic malignancies. Red dots show the variants shared between different hematologic neoplasms or entity in the same patient. Dark blue dots in (A) indicate the variants specially related to chronic myelomonocytic leukemia (CMML), and the black dots in (A) denote the AITL-specific variants. Schematic diagrams depicting hypothetical clonal evolution models of the tumors deriving from mutated hematopoietic stem cells (HSC) are also presented. In patient #5 (A), additional mutations besides the clonal hematopoiesis (CH)-associated mutations were identified and implicated in the disease progression to AITL and CMML, respectively. In patient #20, no additional mutations besides those mutated in HSC are identified. In patient #14, a mutated *EZH2*, indicated by green dot, is implicated in the progression to diffuse large B-cell lymphoma (DLBCL). In all three cases, there are mutations that are shared between the AITL and the concomitant myeloid or B lymphomas, supporting evolution of these neoplasms from a common precursor. PV: polycythemia vena; post-PV PMF, post-PV primary myelofibrosis; ITP: immune thrombocytopenia.

However, the top mutational signature extracted from the non-smokers still matched to the smoking-associated COSMIC Signature 4 (*Figure 2—figure supplement 5*). Although no clear association was demonstrated between smoking and development of AITL/PTCL-NOS, the overall findings suggest undocumented modest SHS as a potential source of the smoking-associated COSMIC Signature 4 seen in the late mutations of AITL/PTCL-NOS. It was estimated that 83.9% of non-smokers in the US population were exposed to SHS to various extents as evidenced by detectable metabolite of nicotine in sera in the early 1990s (*Centers for Disease Control and Prevention (CDC), 2008*). More details are presented in the Discussion section.

## AITL with hematologic neoplasms of other lineages arises from common mutated hematopoietic precursors

Four patients with AITL presented with additional hematologic neoplasms of other lineages. We present here the clonal evolution patterns of these tumors based on the results of the mutation profiling for three patients (*Figure 3*). One of these cases provides genetic evidence for the progression of CH to overt myeloid malignancy through acquisition of additional mutations (patient #5, *Figure 3A*). Patient #20 illustrates that the AITL does not necessarily have to be the initially diagnosed malignancy in patients with both AITL and a second malignancy. The third case was an unusual case in which the patient (#14) had CH, AITL, as well as DLBCL, the latter was associated with acquisition of an *EZH2* hotspot mutation. Detailed descriptions of these illustrative cases are provided in Appendix 1.

Together, our data further provide evidence that AITL can be associated with the development of a hematopoietic neoplasm of different lineages, that is, myeloid or B-lymphoid, either preceding or subsequent to the diagnosis of AITL. In all cases, truncal mutations common to all lineages are seen, with late mutations seen in specific tumors (e.g., *SRSF2* in myeloid, *EZH2* in DLBCL).

## Impact of destructive *TET2* mutations on development of multiple hematologic malignancies

One of the features shared among the four cases with concomitant hematologic neoplasms is that they all had multiple (>1) pathogenic mutations in *TET2*. This observation prompted us to investigate the relationship between *TET2* mutation status and occurrence of multiple hematologic malignancies, specifically through assessing effects of *TET2* mutation status on probability of concomitant hematologic neoplasm-free survival in AITL patients. For Kaplan–Meier analysis shown below, the CHN-free survival time is defined as duration from AITL diagnosis to date of death without CHN or date of last follow-up without CHN (*Supplementary file 1*). The event in the Kaplan–Meier analysis is occurrence of CHN (if yes, 1, no, 0) before they die or the last follow-up (right-censored). To increase the power of the statistical analysis, the patients included in our study were combined with an outside cohort of AITL patients whose relevant genomic and survival/CHN data were recently published (*Lewis et al., 2020*), leading to the total number of 47 cases for CHN-free survival analysis.

The patients were initially divided into two groups: wild-type *TET2* group (no *TET2* mutation found in the BM/PB samples) and pathogenic *TET2* mutant groups (one or more *TET2* mutation detected in the BM/PB samples). Although there was a trend that AITL patients with the pathogenic *TET2* mutations detected in the BM/PB had a worse clinical outcome, no statistically significant differences in the second hematologic malignancy-free survival were observed (p-value=0.3273, stratified hazard ratio [HR] = 0.29), consistent with the literature (*Lemonnier et al., 2012*).

We further stratified the patients into the high *TET2* mutation burden and no or low *TET2* mutation burden subgroups. The criteria for inclusion in the first subgroup (n = 11) are as follows: (1) the CH identified in the BM/PB harbored two or more pathogenic mutations in *TET2*, including pathogenic SNVs, nonsense and frameshift mutations interpreted as 'Tier 1' or 'Tier 2' mutations according to a published professional guideline in molecular pathology (*Li et al., 2017*); (2) the VAF of each of the pathogenic *TET2* variants described in (1) was ≥15% . The cases that did not meet these two criteria were assigned to the second subgroup (n = 36). Kaplan–Meier analysis showed that AITL patients carrying two or more pathogenic *TET2* mutations with high allelic burden (the first subgroup) had significantly shorter time in development of CHN (p-value=0.0006; *Figure 4*). Cox proportional hazards model also estimated a significantly higher HR in the first subgroup (stratified HR, 14.01, 95% CI of ratio, 3.1–63). Further analysis shows that specificity and sensitivity of this CHN biomarker reach 97.2 and 72.7%, respectively, with 88.9% of positive predictive value (PPV) and 92.1% of

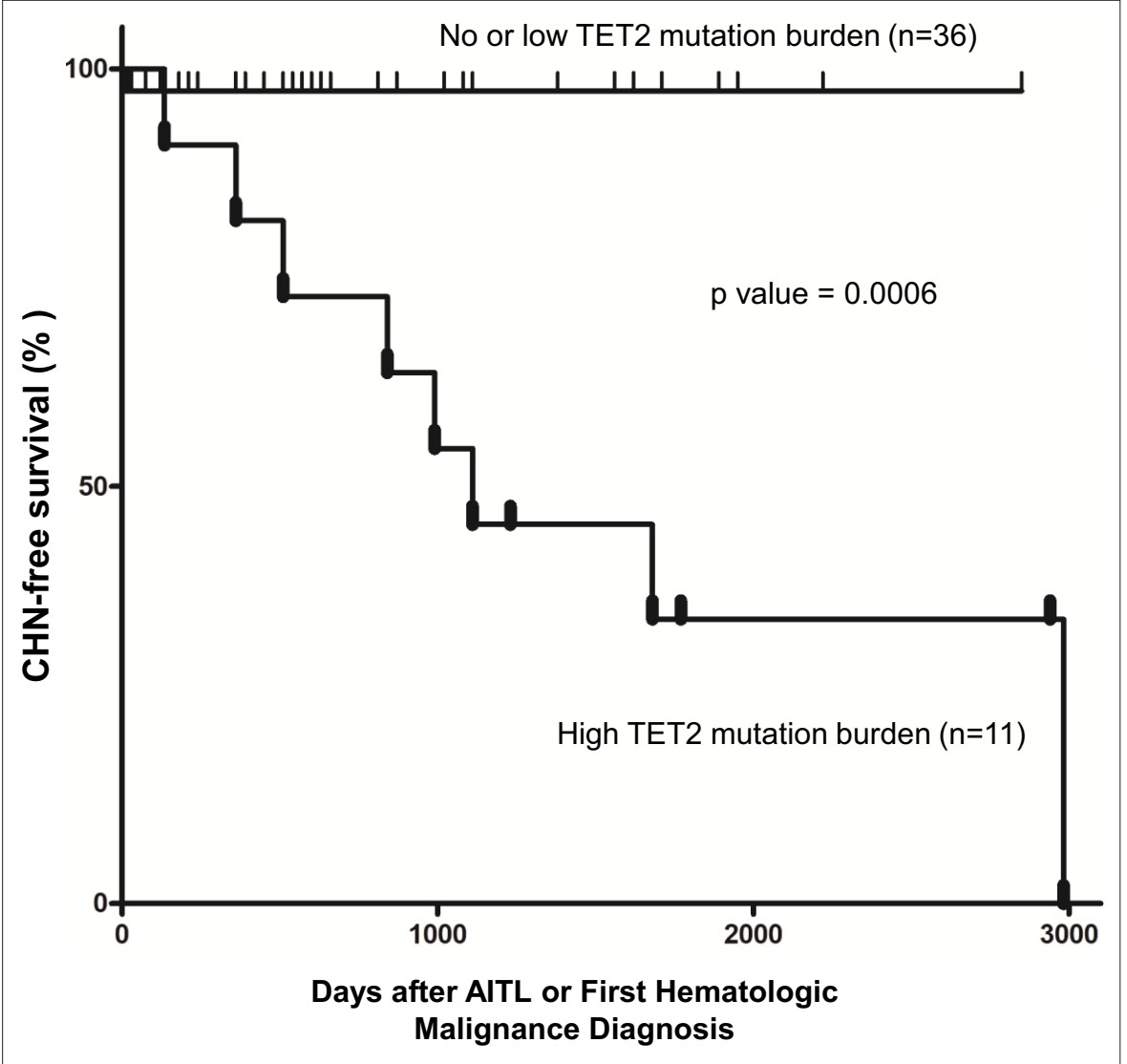

**Figure 4.** Pathogenic *TET2* mutation status in the bone marrow/peripheral blood (BM/PB) samples is a predictive biomarker for concomitant hematologic neoplasms in angioimmunoblastic T-cell lymphoma (AITL) patients. Kaplan–Meier analysis of concomitant hematologic neoplasm-free survival in AITL or AITL-related patients based on *TET2* mutation status in the BM/PB. Concomitant hematologic neoplasm-free survival of AITL patients can be stratified based on absent/low or high TET2 mutation burden subgroups. p-Value was calculated by log-rank test, and p-value<0.05 is considered statistically significant. In one case, the second hematologic malignancy (PV) preceded the development of AITL.

negative predictive value (NPV). This survival analysis indicates that harboring two or more pathological mutations in *TET2* with relatively high allele burden (>15%) is an independent risk factor to predict the second hematologic malignancies in AITL patients.

## Discussion

In the current study, we examined the landscape of the genomic alterations in AITL/PTLC-NOS and their paired BM or PB using a large-panel targeted sequencing approach in the largest cohort of the AITL patients reported to date. We demonstrated that in about 60%  of AITL/PTCL-NOS patients identical pathogenic *TET2* and/or *DNMT3A* mutations were shared between AITL/PTCL-NOS and CH found in the BM or PB. Studies of large cohorts have demonstrated an increased risk of hematologic malignancy for CH (*Genovese et al., 2014*; *Jaiswal et al., 2014*), but no definitive link has been established between CH and AITL/PTCL-NOS from those studies. Our findings suggest that these *TET2* and/or *DNMT3A* mutations may occur very early in the hematopoietic stem cells (HSC) before

they give rise to the common lymphoid progenitors and common myeloid progenitors and propose a possible link between CH and development of AITL (*Fiore et al., 2020a*). Interestingly, the VAF of the CH-associated mutations is 22.5% on average in our cohort and is higher compared to the average VAF of CH-related mutations in the general population (*Genovese et al., 2014*). This observation is in line with the higher risk of hematopoietic malignancy associated with increased VAF (>10%; *Warren and Link, 2020*). It is conceivable that certain *TET2* or *DNMT3A* mutations are stronger drivers that can result in more expanded CH and/or higher efficient T-cell lymphoma development. For example, as seen in patient #28, there were three *TET2* mutations identified in the LN, each of which appears to be present in separate clones and have different capacity to generate CH based on VAF in the BM (0, 5.47, and 10.89%, respectively). In addition, our study supports a mutated HSC as potential origin for AITL. As the *TET2* and/or *DNMT3A* mutations are propagated to the lymphoid and myeloid progeny of the mutated HSC, it can be speculated that in the lymphoid compartment the impacts of these mutations vary depending on the developmental and differentiation stage of the T-cells, and may be most felt in the T-cells of follicular helper cell origin (TFH). Lastly, our interesting case of an AITL patient with CH and subsequent development of DLBCL and the sharing of the same TET2 mutation among all three lesions suggest that a subset of DLBCL, possibly the molecular subtype characterized by mutated *TET2* (*Lacy et al., 2020*), may originate from mutated HSC. To our knowledge, this is the first reported case in which the mutated HSC developed into three distinct tumors of diverse lineages.

The findings from this investigation confirm and extend the results previously published regarding the cellular origin of AITL (*Couronné et al., 2012*; *Lewis et al., 2020*; *Nguyen et al., 2017*; *Quivoron et al., 2011*; *Sakata-Yanagimoto et al., 2014*; *Schwartz et al., 2017*; *Tiacci et al., 2018*). Most of these previous studies presented sporadic AITL cases in which the *TET2* or/and *DNMT3A* mutations present in AITL were also found in their BM/PB compartments. Two reports showed that AITL shared the same *TET2* mutations with the isolated CD20+/CD19+ (B-cells) or CD34+ cells (*Couronné et al., 2012*; *Schwartz et al., 2017*). These studies also pointed to a mutated HSC that gives rise to lymphoid and myeloid cells harboring the same mutations. A high-risk CH was also documented as the cellular origin of AITL and *NPM1*-mutated AML in a patient (*Tiacci et al., 2018*). While our manuscript was under preparation, the results of a study conceptually similar to ours regarding the cellular origin of AITL were reported (*Lewis et al., 2020*). Consistent with our observations, the report showed that the mutations related to CH (i.e., *TET2* or *DNMT3A*) were detected in both the neoplastic T-cell and myeloid compartments in 15 out of 22 AITL patients (68%), and associated with second myeloid neoplasm development after the diagnosis of AITL in 4 cases. However, in their cohort, no cases were reported where AITL developed subsequent to myeloid neoplasms. Our study presented one such case (patient #20) whose AITL developed after 10 years of PV and the two hematologic neoplasms shared three identical *JAK2* and *TET2* mutations (*Figure 3B*). The identification of cases in which myeloid neoplasms precede the diagnosis of AITL provides additional supportive evidence to the postulation that the mutated HSCs are the common origin for these hematologic neoplasms, which develop independently and divergently in tumor evolution. Whether AITL precedes or develops subsequent to the myeloid neoplasms may depend on the stochastic dynamics of the clonal evolution.

Additional late non-CH mutations are found in 68.4% of the AITL/PTCL-NOS in our cohort, consistent with the belief that CH-associated *TET2* and *DNMT3A* mutations are insufficient for tumorigenesis and additional genetic alterations are required. Consistent with this notion, in AITL animal models, *TET2* disruption or *RHOA*$^{G17V}$ expression alone failed to induce AITL development; however, AITL-like lymphoma developed once *TET2* disruption and *RHOA*$^{G17V}$ expression were combined (*Cortes et al., 2018*; *Ng et al., 2018*; *Nguyen et al., 2020*). For the development of myeloid neoplasms, additional mutations beyond CH-associated *TET2* and *DNMT3A* mutations drive further clonal expansion from CH. These mutations may be acquired early (patient #20, *JAK2*, *Figure 3B*) or late during tumor development (patient # 24, *JAK2*, *Figure 1—figure supplement 2*).

We discovered that the late non-CH mutations are enriched for the missense mutations and the C>A substitutions (*Figures 1C and D and 2*). Further analysis identified the major mutational signatures that were very similar (CCS = 0.9) in the Cornell and Kyoto cohorts (LM_Sign.01 vs. Kyoto_Sign.01, *Figure 2—figure supplements 2–4*). The main features shared by these two signatures were the enriched C>A mutations and the closest match with the smoking-associated COSMIC Signature 4 (CCS = 0.5–0.55) among all 30 established signatures (SBS30). The C>A base substitutions as a result of the LM_Sign.01 or Kyoto_Sign.01 signature activity are related to critical mutations in a number of

oncogenic genes, including *RHOA*, *TET2,* and *IDH2*. This finding may have implications on treatment and prevention of AITL. It is believed that the C>A mutations associated with Signature 4 are likely caused by mis-replication or mis-repairing of DNA damage induced by tobacco carcinogens (*Alexandrov et al., 2016*; *Alexandrov et al., 2013*), which largely result in missense mutations (*Blackford et al., 2009*). Consistent with this causative link are our findings that the majority of the late non-CH mutations identified in our cohort were missense mutation (75.2%, *Figure 1D*) and that the patients with AITL/PTCL-NOS have a 172.3-fold increased risk for development of lung cancer compared to the age-matched/adjusted general population (*Figure 2D*). Medical records showed that 19 out of 26 AITL/PTCL-NOS cases were non-smokers, but the major mutational signature extracted from them still matched with smoking-associated Signature 4 (CCS = 0.48) among SBS30 (*Figure 2—figure supplement 5*). This discrepancy may be due to misreporting and undocumented SHS, which were observed in 13.8% of non-smokers with lung cancers (*Alexandrov et al., 2016*). In our cohort, patient #7 was recorded as a non-smoker, but carried many smoking-associated COSMIC Signature 4 mutations (two mutations per megabase, *Figure 2—figure supplement 2C*). The CDC screening study showed that between 1988 and 1994, 20.9% of non-smokers in the US population were exposed to home SHS (at least one family member was a smoker), and 83.9% were exposed to SHS to various degrees during 1988–1994 as cotinine (the main metabolite of nicotine) could be detected at a level of >0.05 ng/ml in the sera of non-smokers. This suggests that most of the patients included in this study might have been exposed to undocumented SHS for ~25–50 years when they were diagnosed with AITL/PTCL-NOS from 2008 to 2019 because 86% of the patients were 50 years old or older (median, 65). Since there is no safe level of SHS exposure (https://www.cdc.gov/tobacco/data_statistics/fact_sheets/secondhand_smoke/health_effects/index.htm), it is conceivable that exposure to undocumented SHS may lead to the gradual accumulation of Signature 4 mutations in the Cornell cohort. In the Kyoto cohort (TFH-PTCL), a similar situation might apply. In Japan, a recent study showed that the overall prevalence of SHS exposure in workplaces, restaurants, and bars were 49, 55, and 83% (*Sansone et al., 2020*). These data may partially explain the accumulation of COSMIC Signature 4-like driver mutations in the non-smokers. Consequently, our findings suggest that cessation of smoking or avoiding exposure to SHS in home or public places may be a potential effective intervention to prevent AITL development in higher risk population, particularly those already found to harbor CH.

On the contrary, the CH-associated genetic alterations in the current cohort are characterized primarily by the C>T and C>G mutations (64% of all the mutations) in *TET2* and *DNMT3A* (*Figure 2A and C*). Analysis revealed that this mutation pattern primarily matched to COSMIC Signature 2 (CCS = 0.65, *Figure 2—figure supplement 1*), which was reported to be associated with the AID/APOBEC family of cytidine deaminases or aging-dependent function decline of base-excision repair machinery (*Alexandrov et al., 2013*). This mutational mechanism might also play a role in the non-CH late mutations as 35% of the non-CH mutations were C>T substitutions (*Figure 2B and C*). Interestingly, reduced accumulation of the C to T mutations by inactivation of *AID* blocked development of B-cell malignancies in aging *TET2*-deficient mice (*Mouly et al., 2018*), implying that AID might be a therapeutic targeting candidate for lymphoma, including AITL.

Furthermore, we found that CH associated with multiple-hit *TET2* (defined as ≥2 pathogenic *TET2* mutations with VAFs of ≥15%) is an independent risk factor for development of concurrent hematologic malignancies in AITL patients (*Figure 4*). Recently, certain features of CH predictive of hematopoietic malignancy development were identified (*Warren and Link, 2020*). These features include >1 mutated gene, VAF >10% , and mutations in specific genes and variants, for example, *TP53* and *IDH1/2*. Our *TET2* biomarker includes two or more mutations and VAF of at least 15% . However, *TET2* has not been previously implicated as a marker for increased risk of hematologic malignancy in CH in general. It is possible that this multiple-hit TET2 biomarker is specific and only relevant in the setting of patients with AITL and CH. Mechanistically, two or more *TET2* mutations each with relatively high mutation burden (≥15%) correlate with increased clonal expansion and/or more severe disruption of TET2 activity, thereby increasing the global chance of acquiring additional driver mutations and hence increased risk for development of second hematologic neoplasms. Consistent with this hypothesis, aging *TET2*-deficient mice develop diverse hematologic malignancies (*Pan et al., 2017*). A reliable predictor for concurrent hematologic malignancies may be helpful for clinical stratification and management for this subset of the AITL patients.

In summary, our study provides genomic evidence of a potential origin of AITL/PTCL-NOS from a mutated HSC clone, which can be associated with CH as well as development of myeloid and even B-cell malignancies. The development of these hematopoietic malignancies of different lineages occurs via divergent evolution from the mutated hematopoietic precursor clone, often with acquisition of additional mutations frequently induced by the C>A-associated mutagenic agents like tobacco. We also identified a potential biomarker: two or more pathogenic TET2 mutations with high mutation burden for the development of second hematologic neoplasm in AITL patients. Single-cell methodology will help definitively determine the clonal architecture in lymphoma and BM with multiple mutations and enhance our understanding of the initiation of tumor induced by CH-associated mutations.

## Additional information

### Funding

| Funder | Grant reference number | Author |
|---|---|---|
| National Cancer Institute | R01 CA194547 | Wayne Tam |

The funders had no role in study design, data collection and interpretation, or the decision to submit the work for publication.

### Author contributions

Shuhua Cheng, Conceptualization, Data curation, Formal analysis, Investigation, Methodology, Project administration, Resources, Software, Validation, Visualization, Writing – original draft, Writing – review and editing; Wei Zhang, Investigation, Methodology, Validation; Giorgio Inghirami, Resources, Writing – review and editing; Wayne Tam, Conceptualization, Data curation, Formal analysis, Funding acquisition, Investigation, Methodology, Project administration, Resources, Supervision, Validation, Writing – original draft, Writing – review and editing

### Author ORCIDs

Wayne Tam (iD) http://orcid.org/0000-0003-4283-0005

### Ethics

Human subjects: This study was conducted in accordance with the Declaration of Helsinki regulations of the protocols approved by the Institutional Review Board of Weill Cornell Medicine, New York, USA. Written consent for use of the samples for research was obtained from patients or their guardians.(#0107004999).

### Decision letter and Author response

Decision letter https://doi.org/10.7554/eLife.66395.sa1
Author response https://doi.org/10.7554/eLife.66395.sa2

## Additional files

### Supplementary files

• Supplementary file 1. Clinicopathological information of patients. AITL: angioimmunoblastic T-cell lymphoma; BM: bone marrow; CH: clonal hematopoiesis; CMML: chronic myelomonocytic leukemia; CTCL: cutaneous T-cell lymphoma; DLBCL: diffuse large B-cell lymphoma; Dx: diagnosis; LN: lymph node; MCF: multi-color flow cytometry; MF: myelofibrosis; MPN: myeloproliferative neoplasm; n/a: data unavailable; PB: peripheral blood; PTCL: peripheral T-cell lymphoma; PV: polycythemia vera; T: type of specimen; TB: estimated neoplastic T-cell burden in BM. &Interval (days) between the lymph node and BM sampling time: negative numbers represent days before diagnosis of AITL or PTCL-NOS by the lymph node biopsy; positive numbers mean days after diagnosis of AITL or PTCL-NOS by the lymph node biopsy. $This TB was estimated based on the limit of detection of the TCRG assay, which is 1–5%.

• Supplementary file 2. The T-cell and myeloid next-generation sequencing (NGS) targeted panels. The gene list shows the genes covered by the T-cell and myeloid targeted panels, respectively. The sequencing summary spreadsheet compares the sequencing performance of the T-cell targeted

panel and the myeloid targeted panel in five bone marrow/peripheral blood (BM/PB) samples.

• Supplementary file 3. Summary of the variants identified by the T-cell targeted panel in angioimmunoblastic T-cell lymphoma/peripheral T-cell lymphoma, not otherwise specified (AITL/PTCL-NOS) and matched bone marrow/peripheral blood (BM/PB). Variant allele frequencies (VAF, %). VAF values highlighted in red represent those attributable to involvement by AITL. VAF values highlighted in blue represent those of the variants present only in the BM or PB. TFH: follicular T helper cell phenotype. C to A base substitution is colored in green. &Detected in diffuse large B-cell lymphoma (DLBCL) (renal tissue) by the myeloid panel.

• Supplementary file 4. Summary of variant types. Mean and median, the average and median number of mutations of the specified subtype per sample, respectively. Percentage, percentage of mutations with the specified subtype relative to the total number of mutations. *A total of 21 bone marrow and 6 PBL samples were sequenced. Among the 27 bone marrow/peripheral blood (BM/PB) samples, 20 had identifiable mutated genes after filtering out the variants due to involvement by the neoplastic T-cells.

• Transparent reporting form

### Data availability
All relevant data are included in this manuscript and the supplementary files.

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

## Appendix 1

### *Figure 1*: six illustrative cases in which common mutated HSC developed into CH and AITL or PTCL-NOS

In patient #1, identical *TET2* and *DNMT3A* somatic mutations, p.E1089fs and p.F791L, were identified both in CH (VAFs: 48.39%, 34.62%) and in the neoplastic T-cells in the diagnostic LN specimen (VAFs: 36.87%, 27.59%) (*Figure 1*). The other three pathogenic variants, *TET2* p.Q743*, *RHOA* p.G17V, and *IDH2* p.R172T, were detected at high VAFs in the LN but at much lower VAFs in the PB (*Figure 1*, *Supplementary file 3*). These results demonstrate that the latter four mutations are likely acquired at a later time point during AITL development (late mutations) and the low-VAF variants detected in the PB represent minimal involvement in the PB and not CH. Patient #4 had no overt evidence of a myeloid neoplasm in the BM while diagnosed with AITL, and had 2.1% BM involvement by AITL according to immunophenotypic and gene rearrangement studies. The T-cell NGS panel identified seven pathogenic mutations in the primary lymphoma, of which six were also found in the matched BM. The allelic burden of the two mutations with the highest VAFs (*DNMT3A* p.Q678*, VAF = 6.22%; *ATP1A3* p.V216M, VAF = 4.47%) was 2–3 times that of the estimated TB (2.1%) in the BM, suggesting that *DNMT3A* p.Q678* and *ATP1A3* p.V216M were not only present in the neoplastic T-cells but also associated with CH. The other five mutations, including *RHOA* p.G17V and *IDH2* p.R172G, were found at the VAFs ranging from 0% to 1.5% in the BM, consistent with lymphoma involvement rather than CH. These results suggest that these five mutations were acquired subsequent to the CH-associated mutations described above. Interestingly, sequencing a relapsed lymphoma specimen from this case identified only one mutation *DNMT3A* p.Q678*. This mutation information in the relapsed lymphoma helps clarify the clonal architecture of both the primary AITL and CH. First, it suggests that the primary AITL harbored two clones, a dominant clone with *ATP1A3*, *DLGAP3*, *TET2*, *RHOA*, and *IDH2* mutations, and a minor clone with *DNMT3A* mutation, the latter being the precursor clone for relapse. Second, it also implies that there are two clones in CH, one associated with the *DNMT3A* mutation and the other with the *ATP1A3* mutation. In patient #10, the CH-associated variant shared between the primary lymphoma and the matched BM, *TET2* p.N1484K, was identified at high allelic burden in both the lymphoma and BM samples (VAFs in LN vs. BM, 47.33% vs. 38.26%). Another *TET2* mutation, p.D1378G, was also identified in both the LN and the BM, at VAFs of 16.89% and 0.32%, respectively. The large difference of the VAF in the BM of the two *TET2* mutations suggests that they may belong to different HSC clones, each of which has markedly different CH contribution, or the *TET2* p.D1378G mutation was acquired subsequent to the *TET2* p.N1484K mutation as a subclonal mutation in the HSC. The other three mutations, *TET2* p.V415fs, *RHOA* p.G17V, and *IDH2* p.R172T, were present exclusively in the lymphoma sample. In patient #29, the *DNMT3A* R882H hotspot mutation was shared at high VAFs in both the lymphoma and BM (46% and 35.6%), consistent with a CH-associated mutation, while the other four mutations detected in the LN were present in the BM at low VAFs, consistent with BM involvement by T-cell lymphoma. Interestingly, both the *TET2* N1484K mutation in patient #10 and the *DNMT3A* R882H mutation in patient #29 were present in the primary lymphomas at a VAF of close to 50%, about 3–4 times the allelic fractions of other mutations identified in the LN. This finding implies the presence of these mutations in almost the entire cell population in the LN. However, based on immuno-morphological evaluation and the VAF of the other mutations identified in the LN, the TB in the LN for these two cases is about 20–40%. This raises the possibility that besides the neoplastic T-cells, reactive lymphocytes in these two cases might also harbor the CH-associated mutations.

In the two PTCL-NOS cases (#2 and #18), we observed similar findings as described above for AITL (*Figure 1*). Patient #2 showed eosinophilia with 1.5% neoplastic T-cells involvement in the BM. Two pathogenic *STAT3* mutations were identified. One could be considered a CH-associated mutation (*STAT3* p.Y657_K658insALL, VAFs in LN vs. BM, 21.61% vs. 8.34%) since its VAF is considerably higher than the estimated percentage of tumor involvement in the BM. The other is most likely PTCL-NOS-related mutation (*STAT3* p.W474*). This *STAT3* nonsense mutation had the allelic burden of 18.33% in the primary lymphoma but was present at much lower level (VAF = 2%) in the matched PB, in line with the estimated TB. Patient #18 presented mildly hypercellular marrow with mild granulocytic hyperplasia while diagnosed with PTCL-NOS and had no involvement in the BM per the overall pathological studies. The NGS target panel identified a splice mutation in *TET2* (c.3501–1G>A) and a missense mutation in *SETX* (p.Y2258D) shared between CH and PTCL-NOS. Two mutations (*TP53* p.P151T and *ARID1A* p.G779*) were only found

in the neoplastic T-cells, presumably representing later mutations in PTCL-NOS development following acquisition of the *TET2* and *SETX* mutations.

## *Figure 3*: three AITL cases with concurrent hematologic neoplasm

Patient #5 was initially diagnosed as AITL, followed by a diagnosis of chronic myelomonocytic leukemia (CMML) 7 months later. In the BM specimen, taken 4 months prior to the diagnosis of CMML and sequenced in this study, the overall pathological findings showed 1–5% involvement by AITL and suspected involvement by a myeloid neoplasm. Three identical mutations, including two *TET2* and one *DNMT3A* mutation, p.I274delinsISfs, p.L1830*, and p.W893S alterations, respectively, were identified in both the LN and BM with high allele burdens (VAF range, 25–50% *Figure 3*, *Supplementary file 3*). In addition, pathogenic *SRSF2* and *JAK2* mutations were identified in the BM. These findings support a scenario in which AITL and CMML arose via divergent evolution from a common HSC clone mutated in the CH-associated *TET2* and *DNMT3A* genes. Subsequent development involves accumulation of additional mutations: *SRSF2* and *JAK2* mutations from CH to CMML, and *IDH2* with other mutations to AITL.

Patient #20 was initially diagnosed with PV and progressed to post-PV PMF after 10 years. Two months later, the patient was also diagnosed with AITL. We sequenced the T-cell lymphoma in the LN and the paired BM specimen with post-PV PMF (the initial PV specimen was unavailable). The overall pathological studies showed no lymphoma involvement in the BM. The NGS revealed that the T- and myeloid malignant cells harbored two identical destructive *TET2* mutations with high mutant allele burdens, p.Q939* (20% VAF) and p. E1026delinsXDfs (42% VAF). The driver mutation for MPN, *JAK2* p.V617F, was not only present in the post-PV PMF (88% VAF), but also found in the concomitant AITL with the mutant allele burden of 54.5% (*Figure 3*, *Supplementary file 3*). No *IDH2* or *RHOA* mutation was detected in the AITL. These findings suggest that the *JAK2* driver mutation was acquired early in the HSC. Indeed, *JAK2* mutation was thought to be an early event in PV (*Bellanné-Chantelot et al., 2006*). In this particular case, the combination of *TET2* and *JAK2* V617F may also be sufficient to drive AITL development as no other drivers like *IDH2* or *RHOA* were found.

Patient #14 initially presented as immune thrombocytopenia (ITP) and had a splenectomy in 2008. He was diagnosed with AITL 2 years later and DLBCL about 7 years later. We sequenced two diagnostic tissue samples for the AITL and DLBCL, respectively, as well as one BM sample collected from this patient in 2010. The morphological, immunophenotyping, and molecular findings confirmed that the BM sample was not involved by AITL or a myeloid neoplasm. Interestingly, two nonsense *TET2* mutations, p.K454* and p.K1799*, were identified in all three samples with the VAFs ranging from 32% to 51%. An additional pathological mutation *EZH2*, p.Y646S, was detected only in the DLBCL (*Figure 3*, *Supplementary file 3*). The results from this patient suggest that the mutated HSC clone, harboring *TET2* p.K454* and p.K1799* alterations, differentiates into progeny cells of three different lineages (myeloid, T lymphoid, B lymphoid), each of which gives rise to CH, or transformed to AITL or DLBCL. The additional *EZH2* mutation has been reported in B-cell lymphomas (*Béguelin et al., 2013*; *Morin et al., 2010*) and likely plays a role in the DLBCL development.

