## [Decision Letter]

**Acceptance summary:**

This paper provides confirmatory data on the previously noted co-occurence of angioimmunoblastic T-cell lymphoma (AITL, a rare T-cell lymphoma characterized by presence of myeloid-like TET2, DNMT3A and IDH2 mutations) and clonal hematopoiesis related to myeloid mutations occurring at the level of hematopoietic stem cell. Through analysis of mutational signatures and observation of frequent secondary lung cancers it further raises a hypothesis of a two-hit pathogenesis of AITL (underlying age-related clonal hematopoiesis followed by smoking-induced driver mutation). It further generates a hypothesis about risk of second hematologic malignancy in patients with AITL related to high-burden TET2 mutations.

**Decision letter after peer review:**

Thank you for submitting your article "Mutation analysis links angioimmunoblastic T-cell lymphoma to clonal hematopoiesis and smoking" for consideration by *eLife*. Your article has been reviewed by 3 peer reviewers, one of whom is a member of our Board of Reviewing Editors, and the evaluation has been overseen by Wafik El-Deiry as the Senior Editor. The following individuals involved in review of your submission have agreed to reveal their identity: Christopher Gibson (Reviewer #1); Philippe Gaulard (Reviewer #2).

The paper was overall reviewed favorably, although the principally novel part relates to the mutation signatures, and that part was also the one most questioned from the methodological point of view. It is not uncertain if this analysis can be made more reliable. Similar questions arose regarding the epidemiologic incidence and survival analyses.

Essential revisions:

1) The analysis of C>A mutation signature (the principal novel finding beyond the Lewis paper) will need to be presented more in-depth to assess its validity which is under question.

– Details of the analytical pipeline used for variant calling and annotation will be needed. The Fiore reference is insufficient.

– DNA changes (apart from protein changes) will need to be listed in the supplemental table 2.

– Coverage data and total read counts for the variants can help alleviate this concern. Discuss strand bias if observed.

– Consider validation in public AITL/PTCL datasets.

2) Association with smoking is undermined by lack of data on smoking history of included patients – please add to the case listing if possible, and report if the C>A signature was observable preferentially in smokers.

3) The fact that origin of mutations (lymphoma, CH, other malignancy) is only hypothesized (on the basis of local VAF) and not confirmed directly is a major weakness, particularly in cases with multiple malignancies. Consider either a more reliable cell-specific confirmation (as performed in one patient), or reframe results and conclusions acknowledging the hypothetical nature of these assignments. The speculation about "mutated hematopoietic stem cells" is very far going without any identification of hematopoietic cells in the samples.

4) The epidemiologic analysis of association between AITL and lung cancer is methodologically questionable. The methods for calculation of "prevalence" of lung cancer in an "aging-matched population" are undefined. Prevalence is an unreliable endpoint as it depends on method/length of observation and is sensitive to ascertainment bias. The validity of statistical testing between very different cohorts shown is highly dubious. Use of standardized incidence rates would be more reliable. Similarly, the analysis of TET2 mutations and secondary cancers is questionable if it does not consider high-dose chemotherapy and differential mortality between the groups, with a suggestion of nearly 100% survival in patients in the "no/low TET mutation burden" which is clinically unrealistic.

5) More information is needed about the cases and samples. The criteria for including 2 PTCL, NOS patients (and not others) are confusing. The temporal origin of the samples (tumor and marrow/blood) is unclear – were these truly "diagnostic" (ie. both before any treatment) or were any after treatment/transplant, and what was the temporal distance between node and marrow sampling.

6) There are many comments on the wording of some conclusions, including unsupported extrapolation of risk from AITL patients to those with age-related CH, unsupported recommendations for treatment – that will need to be corrected and clarified point by point using the "Recommendations for the authors" sections below.

*Reviewer #1 (Recommendations for the authors):*

1. I am concerned that the C>A mutation signature you identify in your late mutations could be partially artifact. C>A (G>T) mutations are a frequent source of artifact in some library preparation methods. Characteristically, they can be identified by bias to the F1R2 read pair configuration (if G>T) or the F2R1 configuration (C>A)(more here: https://gatk.broadinstitute.org/hc/en-us/articles/360035890571-OxoG-oxidative-artifacts). Details of the analytical pipeline used for variant calling and annotation were not provided. OxoG artifacts are usually only a small number of reads, but you did not provide coverage data or total read counts for the variants reported in your supplemental table, so it's not possible to infer the number of alt reads from the VAF.

Additionally, it is not possible to tell which SNVs in the table are C>A since only the amino acid changes are reported. The DNA changes should be reported if one of your findings centers on a mutation signature.

2. As above, smoking data for the patients would be very helpful. Is the C>A signature only observable in smokers? If it is present in non-smokers as well, that would throw the smoking hypothesis into question.

A few comments in the discussion may be overstatements:

– "However, TET2 has not been previously implicated as a marker for increased risk of hematologic malignancy in CH in general."

I would be cautious about extrapolating the risk of HM posed by TET2 mutations from your cohort to the general clonal hematopoiesis population. The patients in your cohort are high-risk by definition since they already have a rare T-cell lymphoma. The fact that multiple TET2 mutations in those patients increases the risk of an additional HM does not necessarily mean that the average CH patient with TET2 mutations has a similar risk.

– "For myeloid malignancies with the double-hit TET2 mutations, more intensive therapeutic regimen like bone marrow transplantation might be warranted for reducing the risk of AITL development."

The overall risk of subsequent AITL could only be determined by examining a cohort of patients with myeloid malignancies. It cannot be extrapolated based on a cohort of AITL patients.

*Reviewer #2 (Recommendations for the authors):*

– The authors should indicate why they include in the study 2 cases of PTCL^-^NOS? On which criteria did they select these 2 cases (among other PTCL-NOS)? Could they detail their main clinical, pathological and phenotypic features? Was a diagnosis of TFH-PTCL excluded?

– The link between APOBEC/AID enriched signature, tobacco, lung cancers and AITL is novel, but relies on a small cohort of AITL patients. Validation on publicly available data of AITL based on exome would be welcome to further assess the enriched association with APOBC/AID activity-associated substitutions at the genetic level and, whenever possible validate in an independent series the link with lung cancers.

– The methodology used to assess the link between CH and AITL/PTCL, ie the histological estimate of the tumor burden in the BM, together with the comparison of the VAFs of the mutated genes, which certainly is conceivable but has some limitations. These limitations should be discussed.

– Why did the authors use a threshold of 15% for the VAF of each pathogenic TET2 variant? It is understandable that this reflect at least in most instances a mutation occurring in early progenitors. Would it be also significant when evaluating the VAF ratio of "early" versus "late" mutations?

– In a recent paper (Blood advance 2021), the prognostic value of DNMT3A mutation or a combination of TET2/DNMT3A and IDH2 mutation was suggested. Could the authors investigate the prognostic relevance in their series?

– CD28 has been previously reported to be mutated in ~10% of AITL. However, this gene mutation is not indicated here? Can the authors explain?

– On page 4, the 18.5% percentage of AITL may be underestimated compared to recent reports

– On page 8, RHOA, CD28, VAV1 are not indicated as "late" mutation?

*Reviewer #3 (Recommendations for the authors):*

1. P6 L83: a more detailed description of the "538-gene targeted NGS panel" (what genes? what sequencing and capture technique? potentially a Supplementary file); the reference to Fiore et al. does not contain information about this assay. It appears that the bone marrow/blood was sequenced using a different assay – what was the overlap or quantitative comparability of these assays?

2. Please indicate (in Supplementary file 1, which would be better fitted in the main paper) the temporal origin of the samples (tumor and marrow/blood) – were these truly "diagnostic" (ie. both before any treatment) or were any after treatment/transplant, and what temporal distance between node and marrow.

3. In 30% of patients there were no CH mutations, although some patients apparently had also typical CH mutations (TET2). Where these cases genomically different than CH-derived AITLs? Are there undetected or regressed CH clones in these patients? (interesting particularly in the context of potential sampling time in relationship to treatment – ie. could CH clones be more/less evident in samples collected after chemotherapy)?

4. There are many figures and tables that just show the same data in the paper, and this would really benefit from a better organization. "Supplementary file 2", "Figure 1—figure supplement 5", "Figure 1—figure supplement 1 C" just show all the same data, including numbers, which are also the same data from Figure 1A, "Figure 1—figure supplement 2", and "Figure 1—figure supplement 3", just in various iterations. I would encourage authors to show these data in a more concise and less repetitive way.

5. The data on different SNV patterns in "CH" and "late" mutations are intriguing, although need validation, and leave some aspects open. The tobacco-related "Signature 4" is characterized by C>A mutations with strand bias – was strand bias observed in these data? Similarly, APOBEC-related C>T mutations should occur preferentially at TpC sites-was this observed?

6. The methods for calculation of "prevalence" of lung cancer in an "aging(sic)-matched population" are unclear and should be explained. Prevalence is a poor indicator of association, as it depends on length of observation and is sensitive to ascertainment bias. The validity of statistical testing between different cohorts shown is highly dubious, as data were not collected consistently. Can the authors use more standard methods (standardized incidence rate) to examine the incidence of lung cancer in patients with AITL compared with truly matched general population? (a cursory look at US registry data suggests that there may be no such association).

7. Data about "evolution" of mutated HSC into AITL or myeloid disorders through acquisition of additional mutations remain speculative without cell separation. In patient 5, there is no evidence that JAK2 variant is "specific to CMML", and it is also unclear if in Patient #14 TET2 variants were present in DLBCL (or AITL for that matter), or were bystanders from CH cells in the tumor. This is difficult to describe as "evidence", and it should be interpreted as an unconfirmed hypothesis, unless the authors can provide some cell type-specific data. The conclusion that "mutated HSC developed into three distinct tumors" is too far-going considering that no attempt to determine presence of mutation is specific lineages was performed.

8. The data on TET2 mutations and secondary hematologic malignancies need to take into context patients' treatment (e.g. receipt of ASCT), and rate of mortality. It is not clear how the endpoint that the authors present was defined-are deaths counted as events? It is unlikely that 80% of studied patients have 100% survival rate. Consider using a more standard method (cumulative incidence) accounting for differential rates of mortality between these groups. Patients who survive for a long time and undergo ASCT have a naturally higher probability of developing a secondary cancer. It is also not clear if patients without evidence of clonal hematopoiesis are included in this analysis, but presence of CH by itself is a known risk factor for hematologic cancers.

---

## [Author Response]

Essential revisions:1) The analysis of C>A mutation signature (the principal novel finding beyond the Lewis paper) will need to be presented more in-depth to assess its validity which is under question.

Thank you, we agree that the C>A mutation pattern should be analyzed more deeply. Five new figures were added as the supplements to Figure 2 to address this concern.

We extracted the major mutational signatures from the genomic data generated in this study (Cornell cohort, late mutations, LM) and a public targeted genomic data set from TFH-PTCL patients (n=44, referring as to Kyoto cohort hereafter) with MutSignatures (a new R package). In these two independent cohorts (Cornell vs Kyoto), the top mutational signatures (LM_Sign.01 vs Kyoto_Sign.01) (new Figure 2—figure supplement 2, 3) were highly active in more than half of the cases in their cohorts, and very similar (cosine similarity ≈ 0.9, new Figure 2—figure supplement 4, p.15-17). Both LM_Sign.01 and Kyoto_Sign.01 were charactered by the enriched C>A mutations at TpCpC trinucleotide motif with transcriptional strand bias (mutation rate of C>A on the forward strand is 2-4 times that of G>T on the reverse strand). These C>A mutations often affected two critical genes (RHOA G17V and TET2 disruption), which is required for development of AITL-like lymphoma in mouse model (PMID: 32704161). Analysis shows that both LM_Sign.01 and Kyoto_Sign.01 have the closest match with smoking-associated COSMIC Signature 4 among all 30 established Signatures (SBS30, version 2), supporting the potential link between AITL/PTCL_NOS and active smoking or secondary exposure to cigarette smoke. More details on the mutational signatures are provided in the main text (p.14-17) or the other replies as shown below.

– Details of the analytical pipeline used for variant calling and annotation will be needed. The Fiore reference is insufficient.

The additional details of the analytical pipeline are added to T Cell Targeted Sequencing at the Methods and Materials section in the revised version of the manuscript (p.8-9).

– DNA changes (apart from protein changes) will need to be listed in the supplemental table 2.

The DNA changes are added to the revised version of Supplementary file 3.

– Coverage data and total read counts for the variants can help alleviate this concern. Discuss strand bias if observed.

The revised Supplementary file 3 includes the sequencing depth and total read counts for each of the variants. As described in the Methods and Materials section (p.9), the variants with potential sequencing strand bias were identified and filtered out by balance ratio relative to read counts, setting as the ratio of 1:5 (balance ratio = the number of forward reads/the number of reverse reads or the number of reverse reads/the number of forward reads). Combined with the overall deep coverage for all the variants called (Supplementary file 3, median coverage per variant = 1366 reads), the ‘fake’ variant calling due to the strand bias in sequencing were ruled out.

– Consider validation in public AITL/PTCL datasets.

As replied above, we analyzed a published targeted sequencing dataset from a large cohort of patients with TFH-PTCL (n=44) (PMID: 31092896). There are several reasons why we chose this public data set: (1) the largest TFH-PTCL sample size published to date; (2) like our T cell panel, the data set was generated by a targeted panel that covered many frequently mutated genes in hematologic neoplasms, including AITL; (3) limited availability of public evaluable exome or whole genome sequencing data in AITL. Analysis showed that the main mutational signature (Kyoto_Sign.01), which was very similar with LM_Sign.01 extracted from the AITL late mutations (CCS ≈0.9), had the closest match with the smoking-associated COSMIC signature 4 (version 2, March 2015), further strengthening the potential link between AITL/TFH-PTCL development and smoking (new Figure 2—figure supplement 2. 3, 4).

2) Association with smoking is undermined by lack of data on smoking history of included patients – please add to the case listing if possible, and report if the C>A signature was observable preferentially in smokers.

Thank you for this excellent point. We added the documented smoking status to the revised Supplementary file 1. Of the evaluable patients (2 patients had no records, and hence excluded), 7 (26.9%) were smokers, 19 (73.1%) non-smokers. Due to statistical limitation, MutSignatures failed to compare de-novo mutational signature from the smokers to the mutational pattern of the non-smokers. There is no significant difference in the C>A or overall mutation burden per sample between these two subgroups (average number of the C>A mutations in the smokers vs nonsmokers: 1 vs 0.92, p = 0.61).

However, the top mutational signature extracted from the non-smokers still matched to smoking-associated COSMIC Signature 4 (new Figure 2—figure supplement 5). For this discrepancy possible explanations are as follows (p.25-26): “This discrepancy may be due to misreporting and undocumented secondhand smoking (SHS), which were observed in 13.8% of lung cancers in non-smokers (PMID: 27811275). In our cohort, Pt#7 was recorded as a non-smoker, but carried many smoking-associated COSMIC Signature 4 mutations (2 mutations per megabase, new Figure 2--figure supplement 2C). The CDC screening studies showed that between 1988 and 1994, 20.9% of non-smokers in the U.S. population were exposed to home SHS (at least one family member was a smoker),and 83.9% were exposed to SHS to various degrees during 1988-1994 as cotinine (themain metabolite of nicotine) could be detected at a level of > 0.05 ng/ml in the sera of non-smokers (PMID: 18614993 and PMID: 30521502). This suggests that most of the patients included in this study may have been exposed to undocumented SHS for more than 25-50 years when they were diagnosed with AITL/PTCL-NOS from 2008 to 2019, because 86% of the patients were 50 years old or older (Median, 65). Since there is nosafe level of SHS exposure (https://www.cdc.gov/tobacco/data_statistics/fact_sheets3/secondhand_smoke/health_effects/index.htm), it is conceivable that exposure to undocumented SHS may lead to the accumulation of Signature 4 mutations in the Cornell cohort. In the Kyoto cohort of TFH-PTCL, a similar situation may apply. In Japan, a recent study showed that the overall prevalence of SHS exposure in workplaces, restaurants, and bars were 49%, 55%, and 83% (PMID: 32033243). These data may partially explain the accumulation of COSMIC Signature 4-like driver mutations in the non-smokers. Consequently, our findings suggest that cessation of smoking or avoiding exposure to SHS in home or public places may be a potential effective intervention to prevent AITL development in higher risk population, particularly those already found to harbor CH.”.

3) The fact that origin of mutations (lymphoma, CH, other malignancy) is only hypothesized (on the basis of local VAF) and not confirmed directly is a major weakness, particularly in cases with multiple malignancies. Consider either a more reliable cell-specific confirmation (as performed in one patient), or reframe results and conclusions acknowledging the hypothetical nature of these assignments. The speculation about "mutated hematopoietic stem cells" is very far going without any identification of hematopoietic cells in the samples.

The paper by the MSKCC team (Lewis et al., Blood advance, 2020) has demonstrated that the shared mutations between CH and AITL were indeed present in CD34+ hematopoietic stem cells. Unfortunately, it is difficult for us to repeat this type of experiment at this moment because we no longer have the archived BM samples for CD34 enrichment. We will certainly consider conducting this type experiment in a future study if suitable samples become available.

As suggested by the reviewer, we have reframed the results and conclusions to reflect the hypothetical nature of these assignments: “Mutation profiling of AITL/PTCL-NOS and matched BM/PB supports a potential origin of AITL/PTCL-NOS from mutated hematopoietic precursors associated with clonal hematopoiesis” (p.11, line 172-174); “Our findings suggest that these TET2 and/or DNMT3A mutations may occur very early in the hematopoietic stem cells (HSC) before they give rise to the common lymphoid progenitors and common myeloid progenitors “ (p.22, line 411-412); “our study supports a mutated HSC as potential origin for AITL” (p.22, line 422-423).

4) The epidemiologic analysis of association between AITL and lung cancer is methodologically questionable. The methods for calculation of "prevalence" of lung cancer in an "aging-matched population" are undefined. Prevalence is an unreliable endpoint as it depends on method/length of observation and is sensitive to ascertainment bias. The validity of statistical testing between very different cohorts shown is highly dubious. Use of standardized incidence rates would be more reliable.

Thank you for this suggestion. We have replaced the prevalence with incidence rates and compared the incidence rates of lung cancer between US general population and the current study cohort: “In these two independent populations, the incidence rates of lung cancer were calculated according to the following formula: new lung cancer /aging-matched population*100000* weight for the age adjustment. Analysis shows that the incidence rate of lung cancer in AITL/PTCL-NOS is 172.3 times higher than that in the age-matched general population (11547 vs 67, P<0.00001) (Figure 2D)” (p.17-18).

The CLL subgroup has been removed because the published data are insufficient for calculation of its incidence rate.

Similarly, the analysis of TET2 mutations and secondary cancers is questionable if it does not consider high-dose chemotherapy and differential mortality between the groups, with a suggestion of nearly 100% survival in patients in the "no/low TET mutation burden" which is clinically unrealistic.

Our interpretation in this section was not very clear. It is indeed impossible that 100% of the AITL patients with no/low TET2 low mutation burden survived. The event in this analysis should have been whether AITL patients developed the concomitant hematologic neoplasm (CHN, if yes, event occurrence) before they died or the last follow-up (right-censored) rather than simply survival or death. The aim is to explore the relationship between TET2 mutations status/burden and occurrence of CHN when the patients live, hopefully to identify a potential biomarker to predict CHN development in this specific population. To clarify this, additional explanation has been added to the text: “This observation prompted us to investigate the relationship between TET2 mutation status and occurrence of multiple hematologic malignancies, specifically through assessing the effects of TET2 mutation status on the probability of concomitant hematologic neoplasm-free survival in AITL patients. […] The event in the Kaplan-Meier analysis is occurrence of CHN (if yes, 1, no, 0) before they die or the last follow-up (right-censored).” (p.19).

Additionally, this analysis has been updated according to the latest CHN status and CHN-free survival time collected on May 20,2021 (Supplementary file 1), the conclusion remains unchanged. We would like to point out that, consistent with the CHN biomarker finding, Pt #24 with high TET2 mutation burden, diagnosed with AITL in June, 2018, developed the concomitant hematologic neoplasm (PV) in November, 2019.

5) More information is needed about the cases and samples. The criteria for including 2 PTCL, NOS patients (and not others) are confusing. The temporal origin of the samples (tumor and marrow/blood) is unclear – were these truly "diagnostic" (i.e. both before any treatment) or were any after treatment/transplant, and what was the temporal distance between node and marrow sampling.

We added the LN and BM/PB sampling times and the recorded treatments to the revised Supplementary file 1 for each of the patients.

The two PTCL cases resembled the AITL cases in that they also harbored CH-associated mutations. Thus, we included these two cases in our study cohort to imply the possibility that our findings may be applicable to broader entities besides AITL.

6) There are many comments on the wording of some conclusions, including unsupported extrapolation of risk from AITL patients to those with age-related CH, unsupported recommendations for treatment – that will need to be corrected and clarified point by point using the "Recommendations for the authors" sections below.

We appreciate these suggestions and have modified the relevant texts accordingly point by point as shown below.

Reviewer #1 (Recommendations for the authors):1. I am concerned that the C>A mutation signature you identify in your late mutations could be partially artifact. C>A (G>T) mutations are a frequent source of artifact in some library preparation methods. Characteristically, they can be identified by bias to the F1R2 read pair configuration (if G>T) or the F2R1 configuration (C>A)(more here: https://gatk.broadinstitute.org/hc/en-us/articles/360035890571-OxoG-oxidative-artifacts). Details of the analytical pipeline used for variant calling and annotation were not provided. OxoG artifacts are usually only a small number of reads, but you did not provide coverage data or total read counts for the variants reported in your supplemental table, so it's not possible to infer the number of alt reads from the VAF.Additionally, it is not possible to tell which SNVs in the table are C>A since only the amino acid changes are reported. The DNA changes should be reported if one of your findings centers on a mutation signature.

Thank you. We have addressed these issues as described above to point #1 of essential revision.

2. As above, smoking data for the patients would be very helpful. Is the C>A signature only observable in smokers? If it is present in non-smokers as well, that would throw the smoking hypothesis into question.

Thank you for this excellent point. Please see our reply to point #2 of essential revision.

A few comments in the discussion may be overstatements:– "However, TET2 has not been previously implicated as a marker for increased risk of hematologic malignancy in CH in general."I would be cautious about extrapolating the risk of HM posed by TET2 mutations from your cohort to the general clonal hematopoiesis population. The patients in your cohort are high-risk by definition since they already have a rare T-cell lymphoma. The fact that multiple TET2 mutations in those patients increases the risk of an additional HM does not necessarily mean that the average CH patient with TET2 mutations has a similar risk.

We share the reviewer’s viewpoint, and our text actually reflected this viewpoint: “However, TET2 has not been previously implicated as a marker for increased risk of hematologic malignancy in CH in general. It is possible that this multiple-hit TET2 biomarker is specific and only relevant in the setting of patients with AITL and CH” (p.27, line 520-522).

– "For myeloid malignancies with the double-hit TET2 mutations, more intensive therapeutic regimen like bone marrow transplantation might be warranted for reducing the risk of AITL development."The overall risk of subsequent AITL could only be determined by examining a cohort of patients with myeloid malignancies. It cannot be extrapolated based on a cohort of AITL patients.

This sentence has been removed.

Reviewer #2 (Recommendations for the authors):– The authors should indicate why they include in the study 2 cases of PTCL-NOS? On which criteria did they select these 2 cases (among other PTCL-NOS)? Could they detail their main clinical, pathological and phenotypic features? Was a diagnosis of TFH-PTCL excluded?

We have addressed this question in the above reply to (5) of essential revision: “The two PTCL cases resemble the AITL cases in that they also harbor CH-associated mutations. Thus, we included these two cases in our study cohort to imply the possibility that our findings may be applicable to broader entities besides AITL.”.

The clinic-pathologic features of these two cases are as follows (p. 6-7):

“Pt. #2: Mesenterial lymphadenopathy found on CT scan during work-up for renal transplant, no morphologic features of AITL, predominantly small cells. The T cells are positive for CD2, CD3, CD5, CD7, CD4, neg for CD8, CD10, BCL6 and PD-1. Diagnosed as PTCL, NOS.

Pt. #18: Abdominal and cervical lymphadenopathy, large pleomorphic cells. The T cells are positive for CD2, CD3, CD5, CD8, TIA-1, granzyme B, TCR alpha-beta, negative for CD7, CD4, CD10, CD56, CD57. Diagnosed as PTCL, NOS, with cytotoxic phenotype”.

Yes, the four patients diagnosed with TFH-PTCL are excluded from two PTCL-NOS cases based on morphological and immunophenotypic features as described on Page 6 (line 90-93): “Of these 27 study cases, 4 were initially diagnosed with PTCL with THF phenotype (Supplementary file 1), and were included in the AITL group based on their similar clinical and molecular features as recently proposed by WHO (Swerdlow et al., 2017)”.

– The link between APOBEC/AID enriched signature, tobacco, lung cancers and AITL is novel, but relies on a small cohort of AITL patients. Validation on publicly available data of AITL based on exome would be welcome to further assess the enriched association with APOBC/AID activity-associated substitutions at the genetic level and, whenever possible validate in an independent series the link with lung cancers.

Thank you for this excellent suggestion, please see our reply to point #1 of essential revision.

For the APOBEC/AID enriched CH mutations in AITL, as far as we know, there is no published genomic data available (for validation) where CH mutations were distinguished from late mutations in AITL patients as the current study did. We will consider this type of analysis in the future once an evaluable published data set is accessible. However, we performed additional signature analysis in the CH mutations (Figure 2—figure supplement 1), further demonstrating the link between the APOBEC/AID-associated variants and AITL as described on Page 14-15: “We extracted two major de-novo mutational signatures (CH_Sign.01 and 02, Figure 2—figure supplement 1A) from the CH-related mutations by MutSignatures (Fantini et al., Scientific reports, 2020). CH_Sign.01 is characterized by the enriched C>T substitutions at the trinucleotide motif Tp**C**pA (mutated base presented as bold), and CH_Sign.02 is enriched with C>T at Cp**C**pA/Gp**C**pG plus the increased C to G substitutions at Tp**C**pG. A cosine correlation similarity (CCS) was used to evaluate closeness between the CH de-novo and COSMIC (SBS30, version 2) signatures. CCS, measured as 1- cosine distance, ranges from zero to one. Zero denotes completely different mutational signatures and one signifies identical signatures. As shown in Figure 2—figure supplement 1B, CH_Sign.01 demonstrates the best match with COSMIC Signature 2 (CCS = 0.65), which is associated with activity of the AID/APOBEC family of cytidine deaminases (Alexandrov et al., 2013). The characterized trinucleotide change in CH¬Sign.01, TpCpA to TpTpA, is also the hallmark of COSMIC Signature 2. Analysis of the CH mutations as the consequence of each mutational signature per sample showed that the activity of CH-Sign.01 dominated in 75% (15/20) of the AITL/PTCL-NOS samples (Figure 2—figure supplement 1C), indicating a potential major role of the AID/APOBEC family of cytidine deaminases in the generation of CH-associated mutations in AITL.”

For the link with lung cancer, we were unable to find evaluable public data for incidence analysis in AITL patients. However, a case report showed that, among concomitant carcinoma, AITL preferred synchronous or metachronous presentation with lung cancer (3 lung cancers out of 8 AITL cases) ( IJICR-2-109.php).

– The methodology used to assess the link between CH and AITL/PTCL, ie the histological estimate of the tumor burden in the BM, together with the comparison of the VAFs of the mutated genes, which certainly is conceivable but has some limitations. These limitations should be discussed.

We agree that there are the limitations for the algorithm used to estimate tumor burdens (TB).

To address this concern, we add one paragraph in the Methods and Materials section as shown on Page 7: “For tumor burden estimate in the BM/PB samples, a complementary strategy was implemented due to limitations of histological or molecular methods. Histological examination has a low sensitivity and AITL cells might lack distinct morphological or immunophenotypic features in the BM/PB samples, potentially leading to false negativity in histological or immunophenotyping estimation in some cases (for example, Pt #1, #5, #12). To avoid these potential pitfalls, besides considering morphological findings, the TB estimate was also based on more objective and sensitive immunophenotypic findings (flow cytometry, Flow). If flow is negative and T-cell receptor γ gene rearrangement (TCRG) is positive, we estimate TB based on the analytic sensitivity of the TCRG assay, which is about 1-5%. If both Flow and TCRG are negative, the VAFs of the T cell lymphoma-associated variants like RHOA p. V17A would be used for estimation by comparison (For example, the PB or BM samples from Pt #1, #22, Supplementary file 1)”.

– Why did the authors use a threshold of 15% for the VAF of each pathogenic TET2 variant? It is understandable that this reflect at least in most instances a mutation occurring in early progenitors. Would it be also significant when evaluating the VAF ratio of "early" versus "late" mutations?

We manually checked the mutation data and found out the potential correlation between the TET2 mutation status/burden (15%) and development of concomitant hematologic neoplasms in the AITL patients, which was confirmed by Kaplan-Meier analysis (Figure 4).

The VAF ratio of the early versus late mutation is an interesting idea, but analyzing our data set did not confirm its potential application in predicting concomitant hematologic neoplasms.

– In a recent paper (Blood advance 2021), the prognostic value of DNMT3A mutation or a combination of TET2/DNMT3A and IDH2 mutation was suggested. Could the authors investigate the prognostic relevance in their series?

We think the reviewer was referring to this paper published in Blood Advances 2021 (PMID: 33496747). This paper proposed that DNMT3A mutation conferred a worse PFS in patients treated with CHOP plus Lenolidomide. However, we cannot do similar analysis in our cohort since the patients were not treated uniformly.

– CD28 has been previously reported to be mutated in ~10% of AITL. However, this gene mutation is not indicated here? Can the authors explain?

In the current study cohort, the CD28 mutation rate was not ~10% but ~4%. The mutation in CD28 was found in one patient (Pt#5, 1 out of 25 AITL cases, Supplementary file 3). In the manuscript, the recurrent mutations were highlighted and presented. Since there was no recurrent mutation in CD28 in our cohort, we did not mention it in the text.

– On page 4, the 18.5% percentage of AITL may be underestimated compared to recent reports.

Thank you. It has been updated to 21-36.1% of PTCLs according to the recently reported data containing the large PTCL patient cohorts (PMID: 26045291; PMID: 32704161).

– On page 8, RHOA, CD28, VAV1 are not indicated as "late" mutation?

RHOA and VAV1 are on Page 13: “The recurrent late mutations were limited to several oncogenes and tumor suppressor genes, including the well-known driver genes like RHOA (67% of the cases), TET2 (48%), IDH2 (33%), PLCG1(10%), TP53(10%), VAV1 (10%)” (p.13, line230-232).

As stated above, CD28 was not shown because it was not recurred in this study cohort.

Reviewer #3 (Recommendations for the authors):1. P6 L83: a more detailed description of the "538-gene targeted NGS panel" (what genes? what sequencing and capture technique? potentially a Supplementary file); the reference to Fiore et al. does not contain information about this assay. It appears that the bone marrow/blood was sequenced using a different assay – what was the overlap or quantitative comparability of these assays?

A new Supplementary file 2 presents the genes covered by the T cell or Myeloid targeted panels. There were 20 overlapped genes between two panels, colored in pink in the table. They are frequently mutated in hematologic neoplasms, including *TET2*, *IDH2* and *DNMT3A*.

For the T cell panel, sequencing library construction and selection were based on hybridization capture method, conducted with KAPA Hyperplus Kit (Roche, Basel, Switzerland), Twist Library Prep Kit (Twist Biosciences, San Francisco, CA, USA), respectively, following the manufactures’ protocols. The libraries were sequenced by Hiseq4000 sequencers (Illumina, San Diego, CA, USA).

More details are described on Page 8: “A 537 gene targeted sequencing panel (Supplementary file 2), based on hybridization capture method for sequencing library construction and selection, were designed to investigate the genomic profile of the primary tumors and the BM/PB tissues (Fiore et al., 2020b). The genomic regions covered by sequencing include coding exons and splice sites of these genes (target region: ~3.2 Mb) that were reported being recurrently mutated (>2) in mature T-cell neoplasms, as well as genomic regions corresponding to recurrent translocations. Using an input of genomic DNA of at least 100 ng isolated from frozen tissues or FFPE samples, the next-generation sequencing (NGS) libraries were constructed using the KAPA Hyperplus Kit (Roche, Basel, Switzerland), and hybrid selection was performed with the probes from the customized Twist Library Prep Kit (Twist Biosciences, San Francisco, CA, USA), according to the manufacturer’s protocols. Multiplexed libraries were sequenced using 150-bp paired end Hiseq4000 sequencers (Illumina, San Diego, CA, USA).”

The Myeloid panel was used to confirm mutations in some BM or PB samples. A spreadsheet, included in Supplementary file 2, summarizes the variants, VAFs and coverages detected by T cell and Myeloid targeted panels. Overall, the results were concordant in the overlapped genes between two assays.

2. Please indicate (in Supplementary file 1, which would be better fitted in the main paper) the temporal origin of the samples (tumor and marrow/blood) – were these truly "diagnostic" (ie. both before any treatment) or were any after treatment/transplant, and what temporal distance between node and marrow.

The temporal origin of the BM/PB samples were added to Supplementary file 1, including reported diagnostic or treatment status while they were sampled.

3. In 30% of patients there were no CH mutations, although some patients apparently had also typical CH mutations (TET2). Where these cases genomically different than CH-derived AITLs? Are there undetected or regressed CH clones in these patients? (interesting particularly in the context of potential sampling time in relationship to treatment – ie could CH clones be more/less evident in samples collected after chemotherapy)?

In the cases with no CH mutations identifiable by the T cell panel (n=8), the genomic mutation pattern was characterized by the enriched *RHOA* (62.5%) and *TET2* (50%) mutations, similar with that of the late mutation in the CH-derived AITLs (*RHOA*, 67%; *TET2*, 48%). Thus there is no difference in major genomic mutation landscape between the cases with and without CH mutations.

Indeed, undetected CH clones in those cases without identifiable CH mutations can’t be ruled out because the T cell panel only covered 537 genes or due to cytotoxic treatment. For example, PPM1D, a gene frequently mutated in CH due to chemotherapy, was not part of the T cell panel.

Additionally, we assessed the effect of chemotherapy on CH mutation burden by comparing the number of CH mutations in the BM/PB samples collected before (n=21) or with ongoing/after (n=6) chemotherapy. For 6 BM/PB samples with chemotherapy, the time interval from start of chemotherapy to BM/PB sampling was 2.5 to 17 months. As shown in (Author response image 1), the number of the CH mutations in such two subgroups was comparable (median: 1 vs 1; mean: 1.52 vs 1.33 per sample), hence the CH mutation burden was similar with and without chemotherapy (p = 0.77, t test).

**Author response image 1. sa2fig1:** 

4. There are many figures and tables that just show the same data in the paper, and this would really benefit from a better organization. "Supplementary file 2", "Figure 1—figure supplement 5", "Figure 1—figure supplement 1 C" just show all the same data, including numbers, which are also the same data from Figure 1A, "Figure 1—figure supplement 2", and "Figure 1—figure supplement 3", just in various iterations. I would encourage authors to show these data in a more concise and less repetitive way.

Thank you. The original "Figure 1—figure supplement 5" and "Figure 1—figure supplement 1 C" have been removed. “Figure 1—figure supplement 2” and “Figure 1—figure supplement 3” are merged as the new Figure 1—figure supplement 2 in the revised version of the manuscript.

5. The data on different SNV patterns in "CH" and "late" mutations are intriguing, although need validation, and leave some aspects open. The tobacco-related "Signature 4" is characterized by C>A mutations with strand bias – was strand bias observed in these data?

As replied to point #1 of essential revision, we observed the strand bias associated with COSMIC Signature 4 (p.16-17). More specifically, of 22 C to A mutations identified in the Cornell cohort, 17 were on the forward strand (C to A) and 5 the reverse strand (G to T). Therefore, the strand bias here is at C:G base pairs where mutation of C exceeded mutation of G by 2.4 folds (17/5=3.4). Consistent with this, among 43 C to A mutations identified in Kyoto cohort (TFH-PTCL), 32 were on the forward strand and 11 the reverse strand (the difference was 1.9 folds). These findings support the similarity among LM_Sign.01, Kyoto_Sign.01 and COSMIC Signature 4 associated with smoking.

Similarly, APOBEC-related C>T mutations should occur preferentially at TpC sites-was this observed?

The answer is Yes. We have provided more details in the reply to point #2 of Reviewer #2.

6. The methods for calculation of "prevalence" of lung cancer in an "aging(sic)-matched population" are unclear and should be explained. Prevalence is a poor indicator of association, as it depends on length of observation and is sensitive to ascertainment bias. The validity of statistical testing between different cohorts shown is highly dubious, as data were not collected consistently. Can the authors use more standard methods (standardized incidence rate) to examine the incidence of lung cancer in patients with AITL compared with truly matched general population? (a cursory look at US registry data suggests that there may be no such association).

Thank you for this suggestion. Please see our reply to point #4 of essential revision.

7. Data about "evolution" of mutated HSC into AITL or myeloid disorders through acquisition of additional mutations remain speculative without cell separation. In patient 5, there is no evidence that JAK2 variant is "specific to CMML", and it is also unclear if in Patient #14 TET2 variants were present in DLBCL (or AITL for that matter), or were bystanders from CH cells in the tumor. This is difficult to describe as "evidence", and it should be interpreted as an unconfirmed hypothesis, unless the authors can provide some cell type-specific data. The conclusion that "mutated HSC developed into three distinct tumors" is too far-going considering that no attempt to determine presence of mutation is specific lineages was performed.

Please see our reply to point #3 of essential revision

8. The data on TET2 mutations and secondary hematologic malignancies need to take into context patients' treatment (e.g. receipt of ASCT), and rate of mortality. It is not clear how the endpoint that the authors present was defined-are deaths counted as events? It is unlikely that 80% of studied patients have 100% survival rate. Consider using a more standard method (cumulative incidence) accounting for differential rates of mortality between these groups. Patients who survive for a long time and undergo ASCT have a naturally higher probability of developing a secondary cancer. It is also not clear if patients without evidence of clonal hematopoiesis are included in this analysis, but presence of CH by itself is a known risk factor for hematologic cancers.

Please see our reply to point #4 of essential revision.